# A comparison of two Stokes ice sheet models applied to the Marine Ice Sheet Model Intercomparison Project for plan view models (MISMIP3d)

Tong Zhang[1,3,4], Stephen Price[2], Lili Ju[4], Wei Leng[5], Julien Brondex[6], Gaël Durand[6], and Olivier Gagliardini[6]

[1]State Key Laboratory of Severe Weather (LASW), Chinese Academy of Meteorological Sciences, Beijing, China
[2]Fluid Dynamics and Solid Mechanics Group, Los Alamos National Laboratory, Los Alamos, NM, USA
[3]State Key Laboratory of Cryospheric Sciences, Chinese Academy of Sciences, Lanzhou, China
[4]Department of Mathematics and Interdisciplinary Mathematics Institute, University of South Carolina, Columbia, SC, USA
[5]State Key Laboratory of Scientific and Engineering Computing, Chinese Academy of Sciences, Beijing, China
[6]Université Grenoble Alpes, CNRS, IRD, IGE, F-38000 Grenoble, France

*Correspondence to:* Stephen Price (sprice@lanl.gov)

**Abstract.**

We present a comparison of the numerics and simulation results for two "full" Stokes ice sheet models, FELIX-S (Leng et al., 2012) and Elmer/Ice (Gagliardini et al., 2013). The models are applied to the Marine Ice Sheet Model Intercomparison Project for planview models (MISMIP3D). For the diagnostic experiment (P75D) the two models give similar results ($<2\%$ difference

5 with respect to along-flow velocities) when using identical geometries and computational meshes, which we interpret as an indication of inherent consistencies and similarities between the two models. For the Stnd, P75S, and P75R prognostic experiments, we find that FELIX-S (Elmer/Ice) grounding lines are relatively more retreated (advanced), results that are consistent with minor differences observed in the diagnostic experiment results and that we show to be due to different choices in the implementation of basal boundary conditions in the two models. While we are not able to argue for the relative favorability of either implementation, we do show that these differences decrease with increasing horizontal (i.e., both along- and across-

10 flow) grid resolution and that grounding line positions for FELIX-S and Elmer/Ice converge to within the estimated truncation error for Elmer/Ice. Stokes model solutions are often treated as an accuracy metric in model intercomparison experiments, but computational cost may not always allow for the use of model resolution within the regime of asymptotic convergence. In this case, we propose that an alternative estimate for the uncertainty in the grounding line position is the span of grounding line

15 positions predicted by multiple Stokes models.

## 1   Introduction

As Earth's largest reservoirs of fresh water, ice sheets are important components of the global climate system. Humans feel their impacts most acutely through changes in global sea level, as ice sheets grow or decay in response to climate forcing and internally controlled dynamics. While the rate of present-day sea-level rise is dominated by ocean steric changes and eustatic

changes due to shrinking mountain glaciers, the eustatic contribution from the large ice sheets (Greenland and Antarctica) has increased in recent decades and is expected to continue increasing in coming decades and centuries (Clark et al., 2015). While currently smaller than the sea-level contribution from mountain glaciers or Greenland, the future sea-level rise contribution from Antarctica is of particular concern; because of inherent dynamic instabilities associated with marine-based ice sheets (see,

e.g., Schoof, 2007a; Schoof and Hewitt, 2013), the Intergovernmental Panel on Climate Change (IPCC) recently highlighted future Antarctic ice sheet evolution as the largest uncertainty with respect to projecting future rates of sea-level rise (IPCC, 2013).

Largely to address these concerns, the international community has focussed intense efforts over the last decade on improving the predictive skill of large-scale, whole-ice-sheet models. These improvements include increased fidelity and accuracy with

respect to the governing nonlinear Stokes-flow equations, increased numerical and computational robustness and efficiency, increased complexity and realism with respect to representation of relevant physical processes, and increased efforts towards partial and full coupling with Earth System Models (e.g., see models described in Cornford et al. (2013); Favier et al. (2014); Seroussi et al. (2014); Feldmann and Levermann (2015); Tezaur et al. (2015)). Alongside and critical to advancing these efforts have been the development of model inter-comparison exercises, which have provided community-based "benchmark"

solutions for gauging the correctness of model output (e.g., Pattyn et al., 2008, 2012, 2013). While designed to be simple, distilling a test for a particular model feature of interest down to it's essence, these exercises are still generally too complicated for the application of formal model verification through the use of analytical or manufactured solutions. Thus, these same model intercomparisons have become increasingly dependent on the output from so-called "full" Stokes ice sheet models – the highest fidelity representation of the equations governing the momentum balance for ice flow – to provide a metric for the

most accurate model solutions available (one example is the Elmer/Ice model (Gagliardini et al., 2013), which has taken part in all of the intercomparison projects referenced above). One clear problem with this practice is that the limited number of Stokes models (often only 1) participating in the intercomparison exercises has prohibited any systematic study of differences in solutions from Stokes models.

Here, we apply a second Stokes model, the FELIX-S model of Leng et al. (2012) (see also Leng et al., 2013, 2014; Zhang

et al., 2015), to the Marine Ice Sheet Model Intercomparison for plan view models (MISMIP3d) experiments (Pattyn et al., 2013). We conduct a careful comparison of the numerical methods used and the solutions produced by FELIX-S and Elmer/Ice. In a recent contribution, Gagliardini et al. (2016) show that both diagnostic and prognostic grounding line (GL) positions from Elmer/Ice exhibit substantial sensitivity as a function of not only the *across-flow* mesh resolution (*along-flow* mesh resolution has been explored and discussed in detail previously, e.g., Durand et al. (2009b)), but also as a function of seemingly arbitrary

choices about how basal boundary conditions are implemented in the model. Here, by "arbitrary" we mean in the sense that it is not obvious if and why one choice should be superior to another. Below, we show that a similar level of sensitivity is apparent when comparing FELIX-S and Elmer/Ice output to marine ice sheet benchmark experiments, even when using the same computational mesh with very high along-flow resolution. While these differences clearly argue for a degree of caution when interpreting Stokes model output as the metric for model solution accuracy, we also show that the differences between

Elmer/Ice and FELIX-S solutions decrease as the mesh resolution increases. The consistency between these two models at

high resolution (e.g., 50 m along flow) lends support for their use as a benchmark for lower fidelity models, provided these benchmark solutions are generated using adequate grid resolution.

The paper proceeds as follows. First, we give a brief overview of the governing Stokes-flow equations for ice flow, which are discretized and solved by Elmer/Ice and FELIX-S. We then discuss in some detail the implementation of boundary conditions – some specific to the problem of simulating marine ice sheets – and how they are implemented in the two models. A brief introduction to the MISMIP3d model setup is then given, followed by a presentation of experimental results for the two models. We then give an in-depth discussion of the similarities and differences between results from the two models, our interpretation of where these differences come from, and an assessment of their significance. We close with summary and concluding remarks.

## 2 Model Description

### 2.1 The Stokes ice flow model

Consider the flow of a viscous, incompressible fluid (ice) in a low-Reynolds number flow. Conservation of linear momentum in expressed by the balance between the stress-tensor divergence and the gravitational body force,

$$\nabla \cdot \boldsymbol{\sigma} = \rho_i \boldsymbol{g}, \tag{1}$$

with $\boldsymbol{\sigma}$ representing the Cauchy (full) stress tensor, $\rho_i$ the density of ice, and $\boldsymbol{g}$ the acceleration due to gravity.

The incompressibility of glacier ice is expressed as

$$\nabla \cdot \boldsymbol{u} = 0, \tag{2}$$

where $\boldsymbol{u} = (u, v, w)$ denotes the ice velocity vector. For glacier ice, the constitutive relation can be expressed as for a Newtonian fluid,

$$\boldsymbol{\tau} = \boldsymbol{\sigma} + p\boldsymbol{I} = 2\eta\dot{\boldsymbol{\epsilon}}, \tag{3}$$

where $\boldsymbol{\tau}$ is the deviatoric stress tensor, $p$ is the isotropic ice pressure, $\boldsymbol{I}$ is the identity tensor, $\dot{\boldsymbol{\epsilon}}$ is the strain rate tensor, and $\eta$ is the "effective" viscosity, defined by Nye's generalization of Glen's flow law (Cuffey and Paterson, 2010) as

$$\eta = \frac{1}{2}A^{-1/n}\dot{\boldsymbol{\epsilon}}_e^{(1-n)/n}. \tag{4}$$

The flow-law exponent $n$ is assigned a value of 3, with $n > 1$ leading to a "shear thinning", non-linear rheology. $A$ is the temperature-dependent rate factor and $\dot{\boldsymbol{\epsilon}}_e$ is the effective strain rate (the square root of the 2nd invariant of the strain-rate tensor).

### 2.2 Boundary conditions

At the upper surface, a stress-free boundary condition applies,

$$\boldsymbol{\sigma} \cdot \boldsymbol{n} = 0, \tag{5}$$

where $n$ is the surface normal vector in a Cartesian reference frame. The lower-ice surface consists of two different boundary conditions (Durand et al., 2009a). For the "grounded" part of the flow where ice is in contact with bedrock (i.e., the ice-bedrock interface), the normal stress exerted by the ice body is larger than ocean water pressure. Here we apply the non-linear friction sliding law prescribed for the MISMIP3d experiments (Pattyn et al., 2013),

$$\sigma_{nt_i} + C|\boldsymbol{u}|^{m-1}\boldsymbol{u}\cdot\boldsymbol{t}_i = 0 \quad (i=1,2), \tag{6}$$

where $C$ is a friction coefficient that is non-zero for grounded ice only, $\boldsymbol{t}$ is the bedrock tangent vector, and $m$ is a friction-law exponent. For the floating part of the flow, where ice is detached from the bedrock (i.e., the ice-ocean interface, or the ice-bedrock interface at minimal floatation), the normal stress exerted by the ice body is smaller than *or equivalent to* the ocean water pressure, we apply a stress balance condition; normal stress, $\sigma_{nn}$, is balanced by the pressure due to buoyancy, $P_w$,

$$-\sigma_{nn} = P_w = \rho_w g(z_w - z), \tag{7}$$

where $z_w$ is the sea level. For the case of $z(x,y) = b(x,y)$, we also need to consider a "contact problem" (Durand et al., 2009a) to decide the actual location of the GL. We discuss the contact problem and its implementation in Elmer/Ice and FELIX in more detail below.

The evolution of both the upper and lower free surfaces are determined by a kinematic boundary condition,

$$\frac{\partial z_i}{\partial t} + u\frac{\partial z_i}{\partial x} + v\frac{\partial z_i}{\partial y} = w + \dot{a}_i, \tag{8}$$

for $i = s, b$ denoting the upper and lower surfaces, respectively, and $\dot{a}_i$ representing the surface or basal mass balance.

Consistent with the MISMIP3d experimental setup, the horizontal velocity is set to $u = 0$ m a$^{-1}$ at the ice divide ($x = 0$ km) and free-slip conditions are applied at the two lateral boundaries ($y = 0$ km and $y = 100$ km). Along the fixed ice shelf front at the downstream end of the model domain, Equations 5 and 7 apply for ice above and below the water line, respectively.

Values for all model constants and parameters, including those that specifically apply to the MISMIP3d experimental protocols, are noted in Table 1.

## 2.3 Lower-order approximations

Lower-order approximations to the full Stokes equations expressed above, such as the "shallow-ice approximation" (SIA; Hutter, 1983) and "shallow-shelf approximation" (SSA; Morland, 1987), come about via geometric scaling arguments. These arguments can be used to show that, for many locations on glaciers and ice sheets, specific gradient terms in the stress and strain-rate tensor expressions above contribute negligibly to the momentum balance. While omitting these terms leads to a significant reduction in the numerical complexity and computational cost involved in solving the momentum balance equations (see e.g., Dukowicz et al., 2010; Schoof and Hindmarsh, 2010), the resulting errors may lead to non-negligible differences in dynamically complex regions of the ice sheet, such as near GLs (Pattyn and Durand, 2013). For this reason, full Stokes models are assumed to provide a better measure of the most complete and accurate solution near GLs, against which solutions from lower-order approximations may be compared in order to assess their accuracy.

## 3 Comparison of Model Numerics

Both FELIX-S and Elmer/Ice discretize the Stokes-flow momentum balance equations using the Finite Element Method (FEM). Both models have undergone extensive formal verification (see Gagliardini et al., 2013; Leng et al., 2013), have been subject to formal convergence studies (see Gagliardini et al., 2013; Leng et al., 2012, 2013), and have been shown to compare very favorably to one another when applied to the Ice Sheet Model Intercomparison for Higher-Order Models (ISMIP-HOM) (Pattyn et al., 2008) experiments (see, Figures 6, 7, 10, 11, 13, and 14 in Leng et al. (2012)). Additional details (and references) for Elmer/Ice are given in Gagliardini and Zwinger (2008) and Gagliardini et al. (2013), and for FELIX-S in Leng et al. (2012, 2013, 2014). Here, we provide a summary of several important similarities and differences between the numerical implementations used by Elmer/Ice and FELIX-S, noting that we view the differences as arbitrary. That is, there are *not* clear arguments for why one choice is superior to another and, in that sense, we view both methods as equally valid.

The first significant difference between Elmer/Ice and FELIX-S is in the choice of finite elements; Elmer/Ice uses hexahedral elements with P1-P1 basis functions (linear in velocity and pressure) and "bubble" function stabilization, whereas FELIX-S uses tetrahedral, Taylor-Hood elements with P2-P1 basis functions (quadratic in velocity, linear in pressure). The second important difference is that Elmer/Ice and FELIX-S use different "masking" schemes for identifying grounded versus floating regions of the lower surface; Elmer/Ice marks the nodes bounding each element whereas FELIX-S marks the element faces. The third important difference, which is a generic FEM implementation issue and not specific to the Stokes-flow problem, is in how the value of the basal friction coefficient, $C$, is applied at the Gaussian quadrature points. FELIX-S calculates the values of $C$ at quadrature points directly (with the accuracy of integration increasing with the number of integration points), whereas Elmer/Ice interpolates the values of $C$ at Gaussian quadrature points from nodal values (Gagliardini et al., 2016) (with the number of integration points needed for a given degree of accuracy determined by the order of the basis function).

In terms of similarities, Elmer/Ice and FELIX-S use the same scheme for evolving the free surfaces, based on an FEM discretization of the kinematic boundary Equation (8) (Gagliardini et al., 2013). The two models also use nearly identical implementations of the contact problem. For FELIX-S, the ocean water buoyancy pressure is compared to the normal *stress* of the ice on the bed and for Elmer/Ice, the ocean water buoyancy pressure is first integrated and then compared to the normal *force* of the ice on the bed (Durand et al., 2009a). While both models solve the contact problem at nodes, the information is used differently; Elmer/Ice uses it to decide if *nodes* in contact with the bed are floating or grounded whereas FELIX-S uses it to decide if nodes in contact with the bed constitute an *element face* that is floating or grounded. We return to the discussion of these different schemes and their impact on model output in Section 6.

Lastly, of the three potentially different ways for defining how the basal friction coefficient $C$ varies over the area of a grounded-to-floating element – "Last Grounded" (LG), "Discontinuous" (DI), and "First Floating" (FF) (discussed in more detail in Gagliardini et al., 2016) – FELIX-S uses what amounts to the DI implementation (the $C$ values are discontinuous across the GL) (Figure 1). However, because the values of $C$ are intimately tied to the location of the GL, and because of the different masking schemes used to decide on grounded versus floating nodes (in Elmer/Ice) or element faces (FELIX-S), a

direct comparison based on the implementation of the friction coefficient is really only meaningful for the P75D (diagnostic) simulation. We also return to this discussion in more detail in Section 6.

## 4   Experimental Setup

We provide a brief review of the MISMIP3d experimental setup, referring the reader to Pattyn et al. (2013) for additional details. Three experiments are conducted and reported on; the "standard" prognostic experiment (Stnd), the prognostic, basal sliding perturbation experiments (P75S and P75R), and the diagnostic experiment (P75D). The Stnd experiment is similar to that conducted in the original, two-dimensional MISMIP experiment for flowline models (Pattyn et al., 2012), where steady-state ice sheet GL positions are examined for a uniform, downward sloping (non-retrograde) bed in the along-flow *(x)* direction, with a uniform basal friction coefficient and uniform bed properties in the across-flow *(y)* direction. The goal is to compare three-dimensional model results to those from the two-dimensional test case, for which analytic solutions are available (Schoof, 2007a). The prognostic P75S experiment starts from the steady-state geometry of the Stnd experiment and introduces a two-dimensional, Gaussian perturbation (a slippery patch) to the basal sliding coefficient field, *C(x,y)*, which introduces changes to the model state (velocity and geometry fields). The ice sheet geometry and GL are then allowed to advance for 100 years. The P75R experiment, which starts from the final state of the ice sheet at the end of the P75S experiment, returns the *C(x,y)* field to its original, uniform distribution, inducing GL retreat. The model is then integrated forward in time for another 100 years. The P75D experiment compares the diagnostic model state when using the P75S geometry calculated by the Elmer/Ice model. Below, we first report on the comparison between Elmer/Ice and FELIX-S for the P75D experiment. We then follow with a comparison for the Stnd, P75S and P75R experiments.

For all experiments (unless otherwise noted) the vertical dimension in both models is discretized with 10 layers. For the P75D and Stnd experiments, the nodal coordinates used by Elmer/Ice and FELIX-S are identical, with along-flow resolution of 50 m in the vicinity of the GL and across-flow resolution of 2500 m. For the P75S and P75D experiments, along-flow resolution is 50 m and across-flow resolution is varied from 2500 to 625 m. In this case the Elmer/Ice and FELIX-S nodal coordinates are not identical, as discussed further below in Section 5.3 (Note that we distinguish identical *nodal coordinates* as distinct from identical *meshes*, because the mesh can also be considered a function of element type, which are different for Elmer/Ice and FELIX-S).

## 5   Results

### 5.1   The diagnostic experiment, P75D

We first compare the two models for the diagnostic experiment, P75D (Figure 2). Both models use the same parameters (e.g., $A$, $C$, and $m$; see also Table 1) and, despite the different element types discussed above, have identical nodal coordinates over the entire model domain. From Figure 2, it is clear that the three velocity components ($u$, $v$ and $w$) for Elmer/Ice and FELIX-S are in close agreement for both the upper and lower surfaces, an indication of inherent consistencies between the

two models. For this experiment, the most direct comparison between Elmer/Ice and FELIX-S is afforded by the DI results as, prior to determining $C$, we directly interpolate the *nodal* basal boundary condition mask from the Elmer/Ice diagnostic solution onto the *element-face* mask used by FELIX-S. In general, for the $x$-component of the horizontal velocity ($u$), the differences are relatively small ($<2\%$) over the entire model domain, relatively less near the ice divide and increase continuously from the GL to the ice shelf portion of the domain (Figure 3). For the $v$ and $w$ velocity components, we observe relatively larger discrepancies in the region of the GL (around km 535 – 555), but still very small differences ($<5\%$) over the majority of the domain (Figure 3).

Despite efforts to make mesh, initial and boundary conditions, and parameter settings identical between the two models, several non-negligible differences discussed above are likely responsible for the small differences in velocities shown in Figure 3. The first likely cause for the small differences is the different boundary masking schemes; as noted above, FELIX-S marks the basal boundary faces in an element-wise manner versus the node-wise manner used by Elmer/Ice. To apply as similar as possible boundary settings for the P75D test case, FELIX-S applies the nodal mask from Elmer/Ice when generating its own element-based mask; element faces in FELIX-S are marked as grounded only if all 3 nodes of a triangle are marked as grounded according to the Elmer/Ice mask. Otherwise, the elements are marked as floating (Figure 1) (We note that this is *not* the same criteria that is used by FELIX-S in the remainder of the experiments to determine the location of floating versus grounded ice, as discussed further below). This may lead to small differences when assembling the element stiffness matrices and the right hand side vectors (for the Dirichlet boundary conditions) as part of the FEM discretization of the Stokes system. Another likely cause for the minor velocity differences in the P75D experiment is the specification of the sliding coefficient $C$ at Gauss quadrature points, as discussed above in Section 3. Finally, despite identical mesh coordinates, Elmer/Ice and FELIX-S use different element types, basis functions, and interpolation schemes as discussed above.

Overall, for the P75D experiment FELIX-S results in larger horizontal velocities ($u$) at the GL than does Elmer/Ice. As a result, FELIX-S exhibits a slightly larger ice flux (1%) through the GL than does Elmer/Ice. This systematic difference between the two models is likely a combination of the different numerical choices discussed above. Again, as these choices appear arbitrary with respect to our current level of understanding, it is not clear that the implementation and results from one model can be distinguished as being superior to the other. In any case, we expect these differences to disappear as the horizontal grid spacing approaches 0 (as discussed below in Section 6).

## 5.2 Stnd prognostic experiment

The comparison of Elmer/Ice and FELIX-S diagnostic experiment results demonstrate that model velocities are within several percent of one another when using identical nodal-mesh coordinates, but that different numerics and/or implementations of boundary conditions result in non-zero differences in the model solutions. In turn, the prognostic experiments demonstrate how those biases accumulate and affect the time-integrated model solutions.

For the Stnd prognostic experiment, FELIX-S uses the same initial ice sheet geometry (based on the boundary-layer theory solution of Schoof (2007b)) and the same along- and across-flow resolution in the vicinity of the GL (50 and 2500 m, respectively) as Elmer/Ice. Moving away from the $\sim$30 km wide region of high resolution near the GL, along-flow mesh resolution

linearly increases to several 10's of km based on a geometric progression. From this initial condition, the forward model is integrated for ∼1300 years, by which time the GL position is close to equilibrium (according to the criteria that the relative rate of volume change is <$10^{-5}$, the same criteria used by Elmer/Ice (Pattyn et al., 2013)). Both models demonstrate a continuous advance of the GL, with FELIX-S reaching a steady state GL position ($x_g$) of 519.85 km (Table 2) and Elmer/Ice reaching

steady state positions of $x_g$ = 529.55, 526.80 and 522.35 km, for LG, DI and FF, respectively (Gagliardini et al., 2016). Apparently, FELIX-S produces a smaller equlibrium-sized ice sheet with a GL position that is several to ∼10 km upstream from that of Elmer/Ice.

We attribute the different equilibrium GL locations to differences in the numerical schemes already discussed above. While the overall retreated grounding line of FELIX-S relative to Elmer/Ice is consistent with the minor velocity differences observed

– FELIX-S produces higher along-flow velocities (and hence flux) upstream from, at, and downstream from the GL, with the time-integrated result of thinner ice (and hence floatation) occurring slightly farther inland relative to Elmer/Ice – we note that Elmer/Ice velocities when using the FF scheme are significantly faster (up to ∼100 m a$^{-1}$ downstream of the GL) than for FELIX-S (Figures 2 and 3), and yet the Elmer/Ice GL when using the FF scheme is still advanced relative to that of FELIX-S (Table 2). Hence, other differences in the two numerical schemes must be more important in contributing to the observed

steady-state GL location differences (we return to the discussion of these differences in greater detail in Section 6). Regardless of the reasons, we note that the differences between the equilibrium positions for the FELIX-S and Elmer/Ice GL locations, for both DI and FF, are very close to or within the range of the estimated truncation error for Elmer/Ice at an along-flow resolution of 50 m in the vicinity of the GL (see Durand et al. (2009a), Figure 6, and Gagliardini et al. (2016), Figure 1c, and related discussions therein). As in Durand et al. (2009a), we find that the modeled equilibrium GL (ice thickness of 685 m) is ∼5 km

upstream of that implied by the floatation condition (ice thickness of 618 m) (Figure 4; compare to Figure 2b in Durand et al. (2009a)).

We repeat the Stnd prognostic experiment with FELIX-S but starting from an initially over-sized configuration, allowing the ice sheet to shrink over time and the GL to retreat to its equilibrium position (as opposed to starting from an under-sized initial configuration with an advancing GL). In this case, an equilibrium GL position is reached after a forward model integration

time of ∼1500 years and we find $x_g$ = 524.50 km, approximately a 5 km difference relative to the case with an advancing GL. This difference in equilibrium GL positions under advanced versus retreated initial configurations is consistent with that found by Durand et al. (2009a) and Gagliardini et al. (2016) and is consistent with a model truncation error of ∼5 km at an along-flow resolution near the GL of 50 m ($\sim 100 \Delta x$). Gagliardini et al. (2016) demonstrated that steady-state GL locations from an advanced or retreated initial condition do converge with increasing grid resolution. Based on the similar truncation

error estimate at 50 m along-flow resolution, and results from the P75S and P75R experiments (discussed next), we speculate the FELIX-S would show similar behavior.

Lastly, we conduct a "qausi-convergence" study for the Stnd experiment by comparing solution error against mesh resolution. In order to control computational costs, the mesh is modified slightly relative to that discussed above. First, the number of vertical layers is reduced from 10 to 5. Second, the "quasi" qualifier indicates that, unlike in a true convergence study, we do

not double the along-flow mesh resolution everywhere in the domain at each step in the study. Rather, the number of across flow

elements is unchanged and resolution doubles *only* over a particular region within the vicinity of the grounding line (based on a geometric progression as in previous work, e.g. Durand et al. (2009a)). Simulations are conducted with along-flow resolution of 1600, 800, 400, 200, 100, and 50 m in this refined region. For the highest along-flow resolution, which coincides with that of the Stnd experiment discussed above, the equilibrium GL position is 519.55 km (a difference of 0.30 km relative to when using 10 vertical layers). Figure 5 shows the Richardson estimate for the solution error versus the along-flow mesh resolution. Slight irregularities in the GL position as a function of increasing resolution result from doubling the mesh resolution in the along-flow direction and in the region of the GL, rather than over the entire mesh. Regardless of these minor irregularities, the GL position is seemingly convergent as a function of resolution with a convergence rate between linear and quadratic. At the finest along-flow resolution of 50 m near the GL, the truncation error estimate is $\sim$300 m ($\sim6\Delta x$).

## 5.3 P75S and P75R prognostic experiments

In the P75S and P75R prognostic experiments, we investigate advance and retreat of the GL following a step-change perturbation in the basal friction distribution, for 100 years, and a return to the initial basal friction distribution, for a further 100 years (the P75S and P75R experiments, respectively), as discussed above in Section 4. The initial condition for the P75S experiment is the steady-state GL position of the Stnd prognostic experiment discussed above. To manage computational costs, especially in experiments where sensitivity to mesh resolution is explored, both models employ regional refinement near the GL. Initial mesh resolution in this region is 50 m along flow near the GL and 2500 m across flow for both models but, because of the different equilibrium GL positions for the Stnd experiment, the area of refined mesh in FELIX-S and Elmer/Ice is located in different regions. Thus, the two meshes have the same refined resolution around the GL but different nodal coordinates for this set of experiments (i.e., the two model meshes are not identical as they are for the P75D experiment). Over the course of the P75S and P75R experiments, the centerline ice thickness at the GL varies by <2% of its equilibrium value reported on in Section 5.2.

Similar to the Stnd experiment, FELIX-S predicts relatively less GL advance (P75S) and / or relatively more GL retreat (P75R) than Elmer/Ice, as shown in Figures 6–8. Similar to Elmer/Ice (Gagliardini et al., 2016), FELIX-S shows a clear sensitivity to the *across-flow* resolution ($\Delta y$); as the number of elements in the $y$ direction increases from 20 to 80 ($\Delta y$ decreases from 2500 m to 625 m), the "reversibility" – i.e. the return to the initial position – of the GL improves (note that we expect >>100 yrs to demonstrate full reversibility (Gagliardini et al., 2016)). More importantly, we also find that as the number of elements in the $y$ direction increases from 20 to 80, the agreement between FELIX-S and Elmer/Ice increases for *all* of Elmer/Ice GL implementations (i.e., LG, DI, and FF; Figures 6–8). For the highest across-flow grid resolution, differences in the FELIX-S and Elmer/Ice DI and FF grounding line position *changes* are close to or below the published truncation error for Elmer/Ice, and differences relative to Elmer/Ice LG are converging to that same value (Figure 9).

## 6 Discussion

As noted above, some fraction of the differences in the prognostic model simulation results can likely be attributed to the small differences in the model velocity fields, as seen in the P75D experiment. In turn, these differences are likely related to the different type of finite elements and basis functions used by the two models. However, we attribute the bulk of the prognostic model simulation differences to differences in the treatment of the contact problem, and more importantly, to the different masking schemes used for the basal boundary conditions.

There are small differences in the way the contact problem is implemented in FELIX-S versus Elmer/Ice; while following the same physical basis for the contact problem, FELIX-S compares the normal stress and the sea water pressure acting at nodes, whereas Elmer/Ice compares the normal and sea water force acting at nodes (Durand et al., 2009a). The result may be that, effectively, Elmer/Ice and FELIX-S "feel" slightly different normal forces (or pressures) at basal nodes of the ice-bed interface, resulting in slight differences when assessing whether a node (Elmer/Ice) or element face (FELIX-S) is grounded or

not. Unfortunately, the different element types used by FELIX-S and Elmer/Ice do not allow for a definitive confirmation of this hypothesis.

  Of greater importance, however, are the different treatments of the basal boundary condition masking schemes discussed in Section 3. Figure 1 provides a schematic summary of the differences in the Elmer/Ice and FELIX-S basal boundary masking schemes and demonstrates how those differences would impact the GL location in the two models for a particular "edge" case.

In the upper part of Figure 1, the nodes marked A and C are unambiguously floating (i.e., $z(x, y, t) > b(x, y)$, so that no contact problem needs to be considered for those nodes). Because FELIX-S considers any element with one or more floating nodes to be floating, elements 3, 4, and 8 are all marked as floating, with the resulting FELIX-S GL position shown by the blue line in Figure 1. For the same geometric configuration, the node-based scheme used by Elmer/Ice defines a slightly different position for the GL, shown by the red line in Figure 1.

In addition to the slightly different grounding line locations, the different basal boundary masking schemes will lead to different profiles for $C$, as shown schematically in the lower part of Figure 1 where we plot approximate nodal $C$ profiles for the two models. These differences come about because, for FELIX-S, the nodal matrix coefficients contain the contributions of $C$ (and other variables) from the surrounding elements. As an example, consider profiles 1 and 3 in Figure 1. The $C = 0$ contributions from elements 3 and 8, and assuming additional floating elements to the north of element 3 and to the south

of element 8, reduce the matrix coefficients associated with the node along the centers of profiles 1 and 3 by a factor of approximately 2/6=1/3 (for these nodes, 2 of the 6 surrounding elements are floating), relative to Elmer/Ice. This estimate is only approximate because, in reality, the nodal coefficients contain additional terms related to the ice velocity and the basis functions, which are not uniform for all elements surrounding a node. Similarly, assuming that additional elements downstream of the FELIX-S GL are also floating, the coefficient at node B along profile 2 will be reduced by ~5/6 relative to the Elmer/Ice

value, since 5 of the 6 surrounding elements are floating.

  We attribute the majority of the differences observed in prognostic model simulations to these slight differences in GL position, and more importantly to these slight differences in the value of $C$. If we again consider profile 2 in Figure 1 (and to a

lesser extent profiles 1 and 3), the relatively larger value of $C$ for Elmer/Ice will lead to relatively less basal sliding there and, eventually, relatively thicker ice. This in turn will make it more likely that neighboring nodes may also eventually ground. The overall, time-integrated result will be that, all other things being equal, the Elmer/Ice masking scheme will favor grounding and/or grounding line advance relative to the FELIX-S scheme. This proposed difference in model behavior is consistent with the differences observed when the two models are applied to the prognostic experiments.

We further note that the differences between the nodal $C$ profiles for Elmer/Ice and FELIX-S shown in Figure 1 are broadly similar to the differences between the DI and FF implementations in Elmer/Ice; despite the DI-like implementation of $C$ in FELIX-S, the different masking scheme results in $C$ values at nodes that effectively "look" more similar to the FF implementation of Elmer/Ice (dashed $C$ profile lines in Figure 1). Simulations using Elmer/Ice with these two different implementation demonstrate differences that are broadly similar to the FELIX-S and Elmer/Ice differences observed here for prognostic simulations; the FELIX-S equilibrium grounding line for the Stnd experiment is closest to that for Elmer/Ice when using the FF implementation (Table 2), the change in FELIX-S GL in the P75S experiment is closest to that observed for Elmer/Ice when using the FF implementation (Table 2), and, at all across-flow resolutions, the advance and retreat curves for FELIX-S in the P75S and P75R experiments most closely resemble those for Elmer/Ice when using FF (Figures 6–8).

Based on our understanding of these model-to-model differences and their hypothesized impact on model simulations, we have a strong expectation that the differences in model outputs will decrease as model resolution increases. As the element size decreases, the differences in ice sheet geometry between nodes – the primary cause for differences in the nodal- versus element-based masking schemes – will also decrease, and the two sets of model results should converge. Indeed this is exactly what we see for the P75S and P75R experiments. Similar to the observation of Gagliardini et al. (2016) that the LG, DI, and FF implementations in Elmer/Ice all converge to a similar solution with increasing resolution, we demonstrate here that the FELIX-S results also appear to converge to that same solution with increasing grid resolution (Figures 6-8). When considering the two most comparable implementations of the basal boundary condition masking schemes (FELIX-S and Elmer/Ice FF), the two models agree for the P75S and P75R experiments to within the estimated truncation error for Elmer/Ice at *all* across-flow grid resolutions explored here (Figure 9). For the Elmer/Ice DI and LG implementations, the differences with FELIX-S as a function of grid resolution are also clearly converging (Figure 9).

The convergence study for the Stnd experiment suggests a GL position error of $\sim 6 \Delta x$ at an along-flow grid resolution of 50 m ($\sim 1/2$ the ice thickness). Conversely, the difference in the GL position for the Stnd experiment when starting from a retreated versus advanced initial condition suggests a GL position error of $\sim 100 \Delta x$ ($\sim 7x$ the ice thickness). The discrepancy between these two possible error estimates suggests that the more conservative of the two truncation error estimates should be used.

## 7 Conclusions

We have conducted a first, detailed comparison of two full Stokes ice sheet models, FELIX-S and Elmer/Ice, applied to the MISMIP3d benchmark experiments. While previous informal comparisons have suggested very close agreement between the

two models (Leng et al., 2012), here we explore the model similarities and differences much more carefully, focussing on how differences in model numerics lead to differences in model outputs when using identical mesh coordinates and forcing, and in particular on differences important for the simulation of marine ice sheet dynamics.

Overall, we find close agreement between the two model outputs for cases where the impact of rather arbitrary choices in the implementation of basal boundary conditions can be minimized; for the P75D experiment, diagnostic solutions (e.g., velocity fields) agree to within $\sim$2-5%. While it is difficult to attribute those small differences to particular numerical choices made by the two models, it is likely that different element types and basis functions and different implementations of the contact problem play a role. More significant differences between the two sets of model results are found for prognostic problems. Overall, we find that equilibrium grounding lines for FELIX-S are relatively more retreated than those for Elmer/Ice (as demonstrated by the Stnd experiment) and that FELIX-S is less inclined to ground, and hence less inclined to show grounding line advance than Elmer/Ice (as demonstrated by the Stnd and P75S and P75R experiments).

A detailed look at the two models strongly argues that differences in the basal boundary masking schemes and in the implementation of the basal friction coefficient are the source of these differences. As we are currently unable to judge whether or not one scheme is superior to the other, our results urge caution when interpreting the results from full Stokes models as a metric for accuracy in model intercomparisons, particularly if those results are not obtained at grid resolutions demonstrated to be within the regime of asymptotic convergence. In cases where an estimate for the model truncation error is not available (e.g., due to model cost with increasing resolution), we propose that an alternative estimate for the uncertainty in the grounding line position is the span of grounding line positions predicted by multiple Stokes models. Here, we are encouraged to find that, (1) as the grid resolution for both models increases the differences between the two models continues to decrease, and (2) for their most comparable implementations, the models agree to within the estimated truncation error for one of the models. This finding suggests (but does not prove) that, in the limit of high grid resolution, multiple full Stokes models can be shown to agree on a particular test case solution, despite small differences in their numerics.

Future efforts could improve on the work presented here by confirming the truncation error for the FELIX-S model, in order to understand if different numerics might be a means for further reducing model truncation error. Also, by running simulations at even finer grid resolutions, future efforts could definitively confirm that the results from multiple Stokes models converge in the limit of very fine grid resolution.

*Author contributions.* TZ and SP initiated the study with input from OG, GD, and JB. Necessary modifications to the FELIX-S model were made by TZ with input and guidance from original code authors LJ and WL. TZ conducted the FELIX-S simulations. JB, OG, and GD conducted Elmer/Ice simulations, provided results for comparison with FELIX-S, and provided insight when attributing simulation differences to model differences. TZ and SP wrote the paper with contributions from all co-authors.

*Acknowledgements.* The authors thank Steph Cornford, three anonymous reviewers, and the editor Hilmar Gudmundsson for suggestions that helped to clarify and improve the paper. Support for TZ, SP, LJ, and WL was provided through the Scientific Discovery through Advanced Computing (SciDAC) program funded by the U.S. Department of Energy (DOE), Office of Science, Advanced Scientific Computing

Research and Biological and Environmental Research. TZ was also supported by the National Basic Research Program (973) of China under grant No. 2013CBA01804. LJ was partially supported by the US National Science Foundation under grant No. DMS-1215659. WL was partially supported by the National 863 Project of China under grant No. 2012AA01A309 and the National Center for Mathematics and Interdisciplinary Sciences of the Chinese Academy of Sciences. Elmer/Ice development and simulations presented here were partly funded by the Agence Nationale pour la Recherche (ANR) through the SUMER, Blanc SIMI 6-2012. FELIX-S simulations presented here used computing resources of the National Energy Research Scientific Computing Center (NERSC; supported by the Office of Science of the U.S. Department of Energy under Contract DE-AC02-05CH11231). Elmer/Ice simulations discussed in this paper used computing resources of CINES (Centre Informatique National de l'Enseignement Supérieur, France) under allocations 2015-016066 made by GENCI (Grand Equipement National de Calcul Intensif). This study was inspired by discussions with Frank Pattyn and Gaël Durand at the first MISOMIP workshop, supported by the Center for Global Sea-Level Change at New York University Abu Dhabi.

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

**Tables and figures**

**Table 1.** Parameters used in Elmer/Ice and FELIX-S.

| Symbol | Constant | Value/Units |
|--------|----------|-------------|
| $\rho_i$ | ice density | 900 kg m$^{-3}$ |
| $\rho_w$ | water density | 1000 kg m$^{-3}$ |
| $g$ | gravitational acceleration | 9.8 m s$^{-2}$ |
| $n$ | flow law exponent | 3 |
| $A$ | flow law parameter | $10^{-25}$ s$^{-1}$Pa$^{-3}$ |
| $C$ | Bed friction parameter | $10^7$ Pa m$^{-1/3}$ s$^{1/3}$ |
| $m$ | Bed friction exponent | 1/3 |
| $\dot{a}_s$ | Accumulation rate | 0.5 m a$^{-1}$ |

**Table 2.** Comparison between Elmer/Ice (LG, DI and FF) and FELIX-S GL positions for the Stnd and P75S experiments. The $x_{G_0}$ denotes the steady state GL position for the Stnd experiment. The rows for $\Delta x_{GLc}$ and $\Delta x_{GLm}$ denote the differences between $x_{G_0}$ and the GL position at year 100 in the P75S experiment, at the centerline and margin, respectively. As it is invariant in the across-flow direction, we do not explore the sensitivity of the Stnd experiment to across-flow resolution. All GL positions and differences are given in km.

| | FELIX-S | | | Elmer/Ice (LG) | | | Elmer/Ice (DI) | | | Elmer/Ice (FF) | | |
|---|---|---|---|---|---|---|---|---|---|---|---|---|
| $N_y$ | 20 | 40 | 80 | 20 | 40 | 80 | 20 | 40 | 80 | 20 | 40 | 80 |
| $x_{G_0}$ | 519.850 | – | – | 529.550 | – | – | 526.800 | – | – | 522.350 | – | – |
| $\Delta x_{GLc}$ | 0.100 | 4.350 | 9.400 | 18.950 | 16.350 | 15.050 | 9.250 | 10.825 | 11.950 | 1.950 | 6.425 | 9.900 |
| $\Delta x_{GLm}$ | −14.050 | −8.950 | −6.250 | −0.100 | −2.750 | −3.850 | −8.000 | −7.050 | −6.250 | −13.050 | −10.250 | −7.850 |

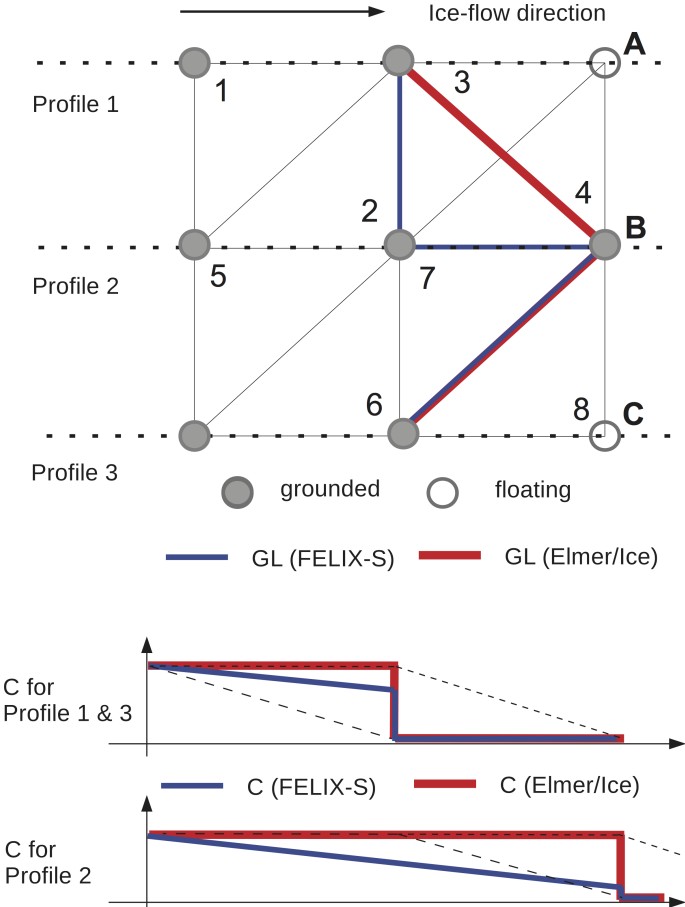

**Figure 1.** A schematic of the different basal boundary masking schemes used by FELIX-S and Elmer/Ice and their impact on the definition of the MISMIP3d basal friction coefficient ($C(x,y)$ is assumed uniform beneath grounded ice for illustrative purposes. For floating ice, $C(x,y) = 0$.). Circles denote the nodes at the ice-bed interface, defining the basal finite element faces (triangular and quadrilateral for FELIX-S and Elmer/Ice, respectively); open circles denote floating nodes for which $z(x,y,t) > b(x,y)$ and solid circles denote grounded nodes for which $z(x,y,t) = b(x,y)$ and $-\sigma_{nn} > P_w$. Numbers 1-8 identify triangular element faces of FELIX-S and letters A-C identify specific nodes common to both models. As discussed in Section 6, the different masking schemes lead to the different grounding line positions and also to the different nodal values of $C$ along profiles 1-3. The $C$ profiles based on DI for Elmer/Ice (heavy red line) and FELIX-S profiles (heavy blue line) are shown, as are the corresponding Elmer/Ice FF (black dashed) and LG (black dotted) profiles.

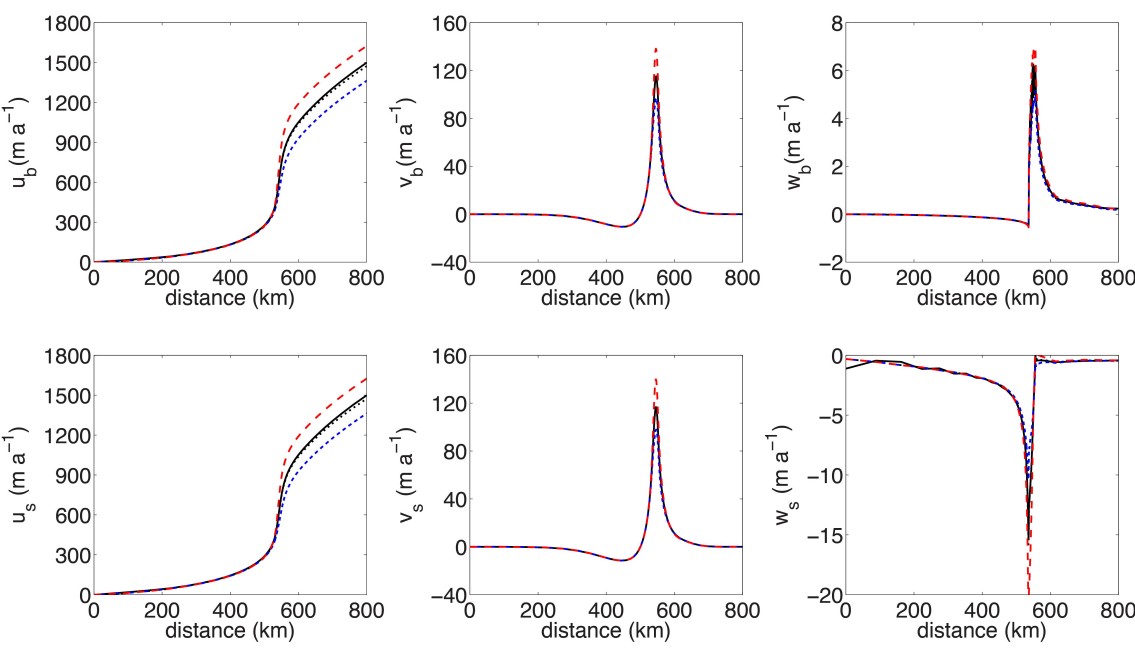

**Figure 2.** Comparisons of mean across-flow velocities for lower ($u_b$, $v_b$, $w_b$) and upper ($u_s$, $v_s$ and $w_s$) surfaces along the $x$ direction for FELIX-S (black-solid line), Elmer/Ice FF (red-dashed line), DI (black-dotted line) and LG (blue-dotted line) cases for the diagnostic experiment P75D. Where the black dotted line is not clearly visible, Elmer/Ice and FELIX-S solutions are overlying.

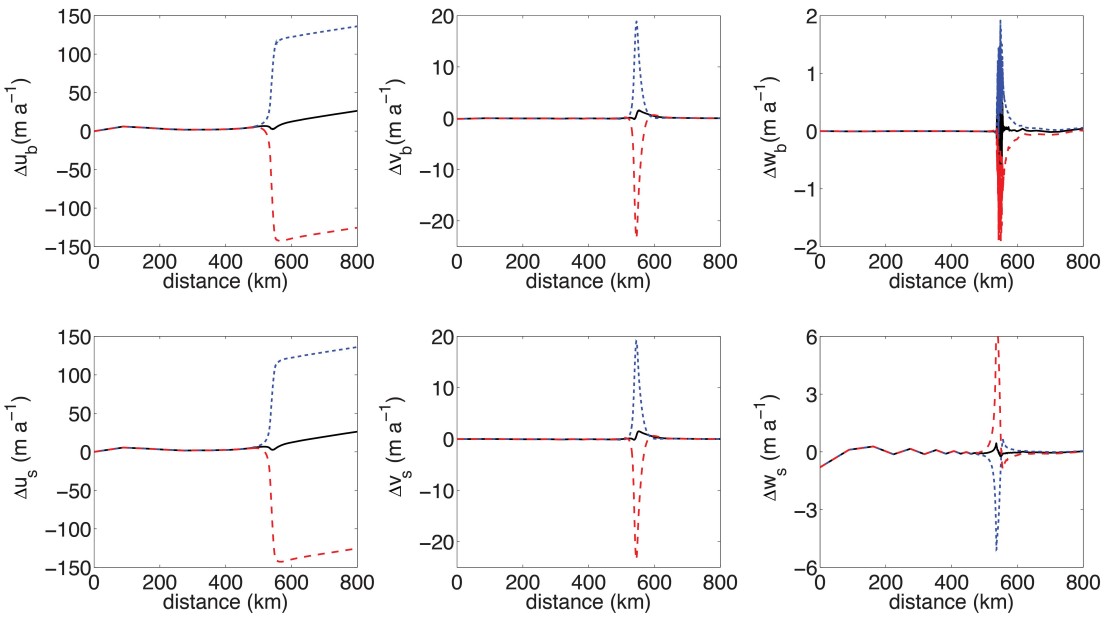

**Figure 3.** Comparisons of mean across-flow velocity differences for lower ($\Delta u_b$, $\Delta v_b$ and $\Delta w_b$) and upper ($\Delta u_s$, $\Delta v_s$ and $\Delta w_s$) surfaces along the $x$ direction for FELIX-S and Elmer/Ice for the diagnostic experiment P75D. The blue-dotted, black-solid, and red-dashed lines denote the differences by substracting Elmer/Ice LG, DI and FF values from FELIX-S values, respectively.

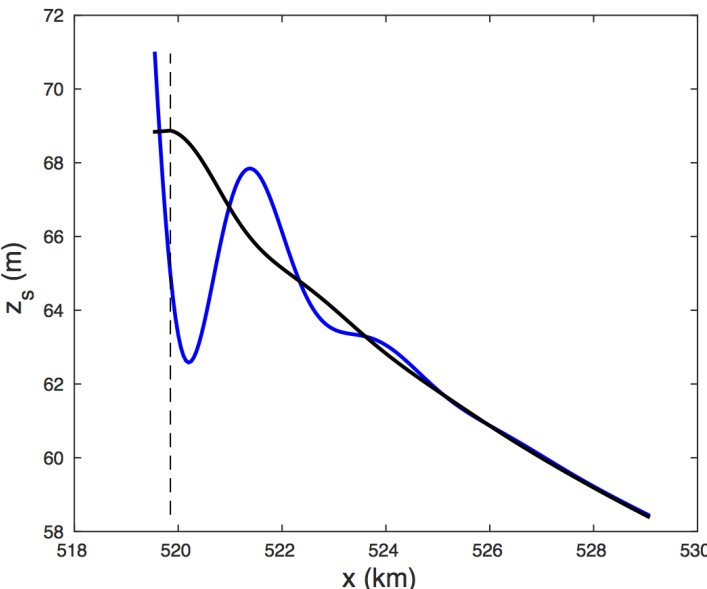

**Figure 4.** Grounding line position versus local floatation. The vertical black-dashed line marks the equilibrium GL position for the Stnd experiment and the heavy blue and black lines denote the modeled ice sheet surface near the GL (blue) versus that determined by the floatation condition (black) (Compare with the inset in Figure 2b from Durand et al. (2009a)).

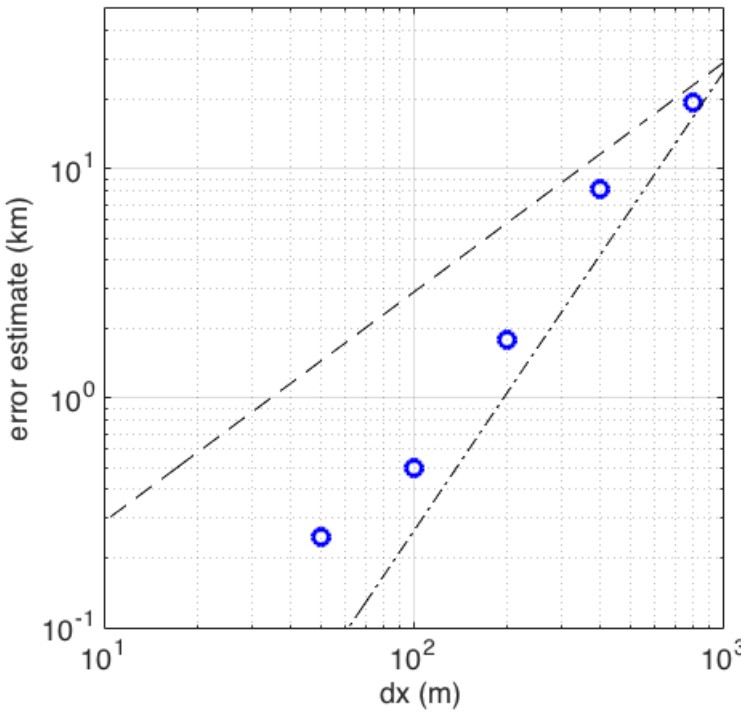

**Figure 5.** Convergence of the Stnd experiment as a function of along-flow grid resolution (circles), as discussed in Section 5.2. Error estimate for grounding line position are based on Richardson error estimation. Black-dashed and dash-dot lines show perfect linear and quadratic convergence rates (respectively).

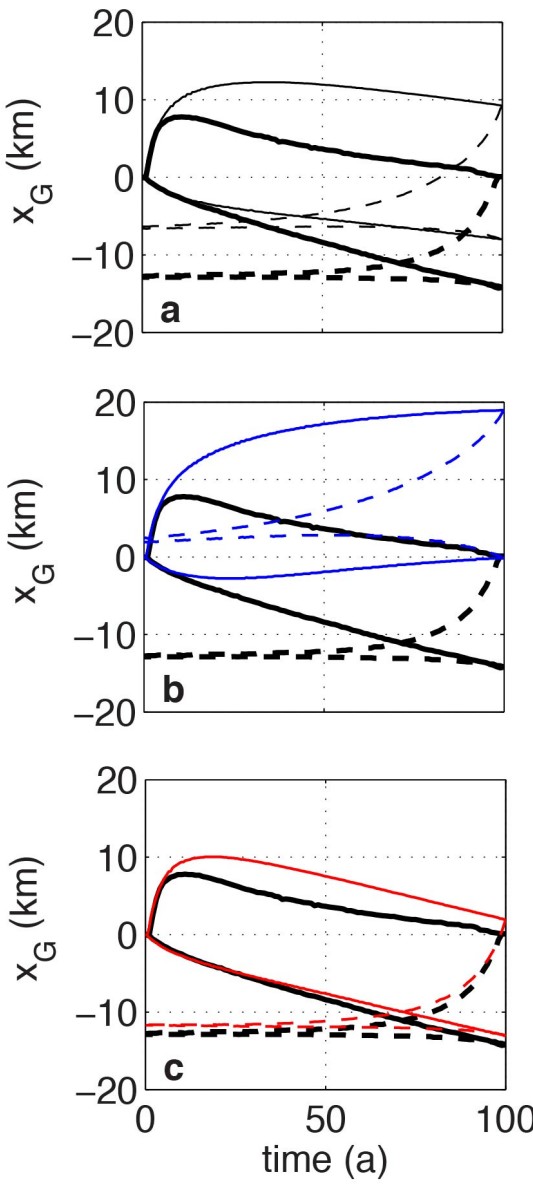

**Figure 6.** GL evolution on both the symmetry axis (upper curves) and free-slip boundary (lower curves) for the P75S (solid curves) and P75R (dashed curves) comparisons between FELIX-S (bold-black curves) and Elmer/Ice DI (a; thin-black curves), LG (b; thin blue curves) and FF (c; thin red curves). The number of elements along the $y$ direction is 20 ($\Delta y = 2500$ m). Note that GL positions are plotted relative to their equilibrium positions in the Stnd experiment.

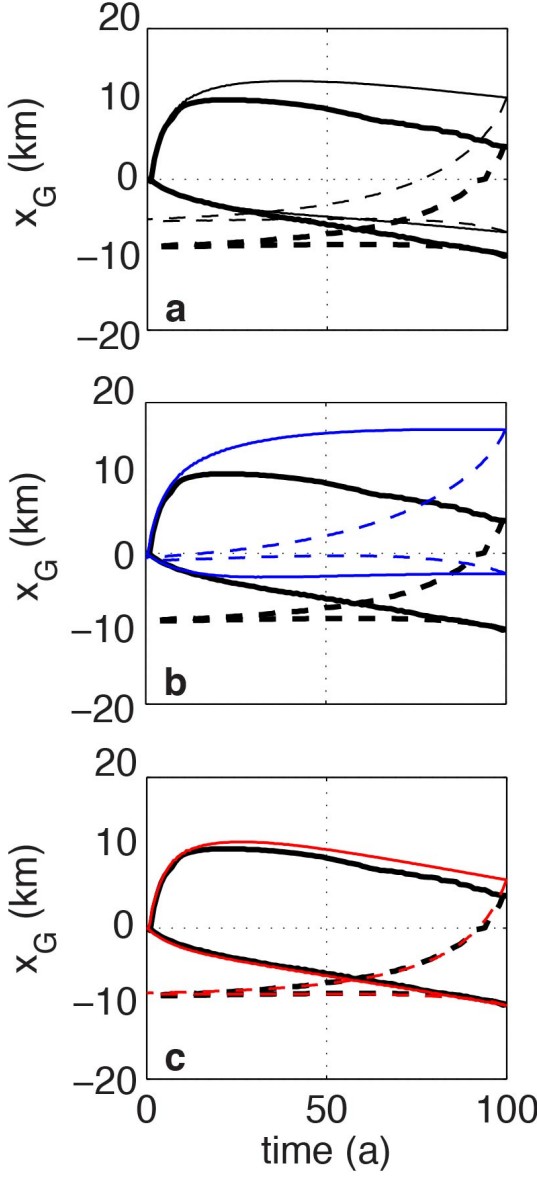

**Figure 7.** GL evolution on both the symmetry axis (upper curves) and free-slip boundary (lower curves) for the P75S (solid curves) and P75R (dashed curves) comparisons between FELIX-S (bold-black curves) and Elmer/Ice DI (a; thin-black curves), LG (b; thin blue curves) and FF (c; thin red curves). The number of elements along the $y$ direction is 40 ($\Delta y = 1250$ m). Note that GL positions are plotted relative to their equilibrium positions in the Stnd experiment.

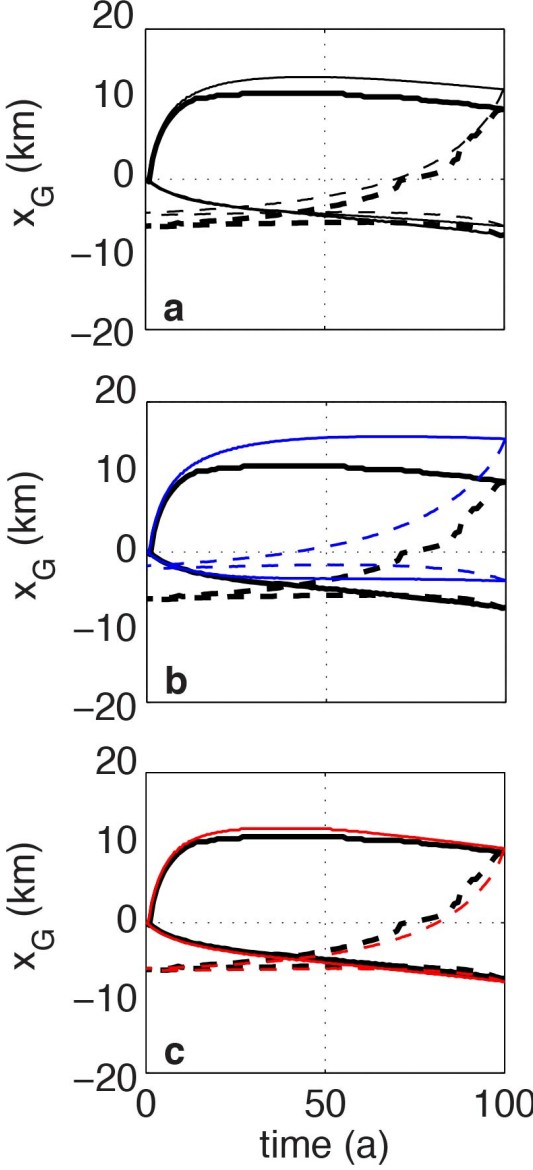

**Figure 8.** GL evolution on both the symmetry axis (upper curves) and free-slip boundary (lower curves) for the P75S (solid curves) and P75R (dashed curves) comparisons between FELIX-S (bold-black curves) and Elmer/Ice DI (a; thin-black curves), LG (b; thin blue curves) and FF (c; thin red curves). The number of elements along the $y$ direction is 80 ($\Delta y = 625$ m). Note that GL positions are plotted relative to their equilibrium positions in the Stnd experiment.

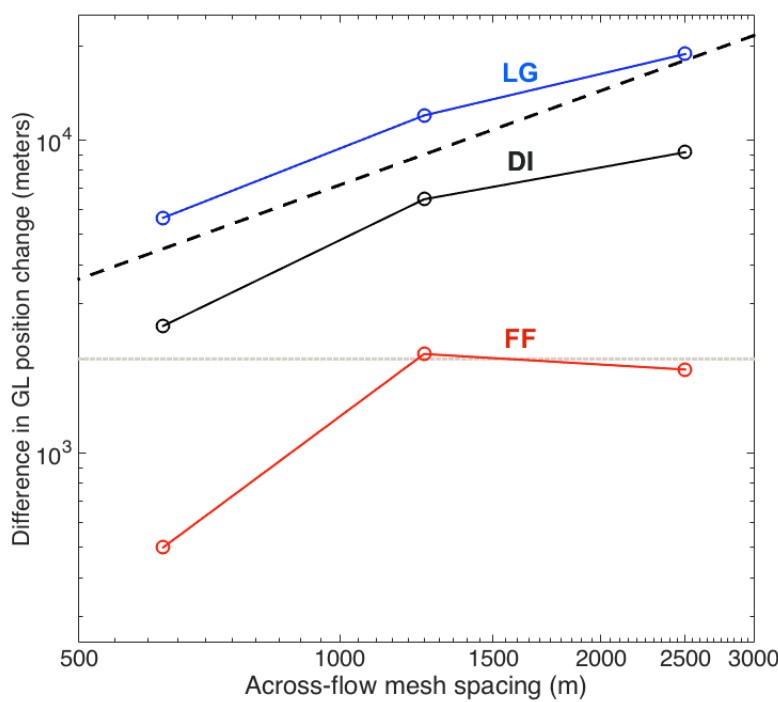

**Figure 9.** Difference in FELIX-S and Elmer/Ice GL position changes ($\Delta GL_{Elmer}$ - $\Delta GL_{FELIX}$) at the centerline for the P75S experiment as a function of increasing across-flow ($y$) resolution (resolution increases from 2500-625 m as $N_y$ increases from 20-80). Lines representing the differences relative to the LG, DI, and FF implementations in Elmer/Ice are labeled. Black-dashed line shows the slope for a theoretical first-order convergence rate. Grey-dashed line shows the estimated Elmer/Ice truncation error of $\sim$2 km, from Durand et al. (2009a) and Gagliardini et al. (2016).