# Peer review of "A comparison of two Stokes ice sheet models applied to the Marine Ice Sheet Model Intercomparison Project for plan view models (MISMIP3d)"

_The Cryosphere, 2016_

## Referee Comment (RC1) · Anonymous Referee #1 · 29 Mar 2016

**1   General statement**

The manuscript "A comparison of two Stokes ice sheet models applied to the Marine Ice Sheet Model Intercomparison Project for plan view models (MISMIP3d)" by T. Zhang compares the results of two finite element full-Stokes models on a widely used benchmark designed to assess the accuracy of models to represent grounding line evolution. It describes the differences in the numerical implementation of the two models: type of elements, friction applied around the grounding line, choice of floating and grounded areas. The manuscript concludes on the importance of including more than one full-Stokes model in intercomparison projects to provide a measure of their

uncertainty. The paper is clear, well-written, and the figures usually appropriate. However, the conclusions presented do not accurately reflects the results presented in this manuscript.

I have several major concerns with the manuscript. First, I am questioning the novelty of this paper. The full-Stokes treatment of grounding lines has been investigated for almost 10 years, and a recent paper by Gagliardini et al. (2016) discusses the problem of numerical convergence due to the treatment of basal friction. The only new element introduced here is the development of grounding lines within FELIX-S, which I believe would be a better fit for Geoscientific Model Development.

Second, I found the abstract in particular to be misleading, and to put emphasis on the "wrong" solutions to real questions. As mentioned by the authors, "as grid resolution increases the grounding line positions for FELIX-S and Elmer/Ice appear to converge", which suggests that full-Stokes models (just like any numerical model) should do a convergence study to assess the impact of grid resolution on their results. The authors on the opposite suggest that "future model intercomparisons using full-Stokes models as a metric should include more than one model, to provide both additional confidence in the results from full-Stokes models and a measure of their uncertainty". Comparing two different full-Stokes models will not allow quantifying the error in any of these models. The only way to assess the error of each model is to do a convergence study, with varying grid resolution, to see if the results are converging, and if so, to quantify the error associated to the discretization. To my knowledge, this has never been properly done in three-dimensional models, and needs to be done instead of writing numerous papers on variations along the same subject (comparison of friction treatment around the grounding line, comparison of several full-Stokes models, ...).

Another point that I think should be emphasized in this manuscript is that not only is the resolution in the along-flow direction important, as noted in many previous studies (Durand et al., 2009; Pattyn et al., 2012; Pattyn et al., 2013), but also the across flow resolution matters for grounding line advance and retreat. So instead of just mentioning the difference between the two models is reduced for increased "grid resolution" (currently what is said in both the abstract and the conclusions), the authors should clearly state that across flow resolution is also an important factor, which has not been really considered in the previous studies.

Finally, the results presented in this paper do not present major new results, as the conclusions are somewhat similar those of Gagliardini et al. (2016), which already discuss the impact of different treatment of basal friction around the grounding line. As errors in full-Stokes models start to be better estimated, and lower order approximations models also improved grounding line treatment during the past few years (Feldmann et al., 2014; Seroussi et al., 2014), a more interesting study would be to reassess the difference between full-Stokes and lower order approximations based on all these new data.

**2 Specific comments**

The convergence or absence or convergence with grid resolution should be rigorously assessed using a convergence study instead of providing statements like "appear to converge" in the abstract.

The description of the friction coefficient in section 3 is not clear. Figure 1 is also confusing, especially the friction coefficient along profile 2. This seems to be a straight line while there is a node in the middle of the profile. The details provided in the discussion should be moved in this section (p.5 l.15-16).

p.8 l.3-7: For which model is there a 5 km difference? This paragraph also mentions that this distance is expected to decrease with increased grid resolution. Why not do these runs and provide an accurate answer? These positions should be the same for a model that has converged with grid resolution. So here again, this seems to suggest that the results presented have not converged. How long is the steady-state run for?

Could this difference be partly explained because the steady-state should be run longer to fully converge?

p.8 l.22-23: "the agreement between FELIX-S and Elmer/Ice increases for all of Elmer/Ice GL implementations". This seems like another sign that the differences in the results are caused by the non-converged aspect of the results presented.

**3 Technical comments**

p.1 l.2 (and other places in the text): full Stokes is commonly used in the literature, so there is no need to use "full" Stokes.

p.1 l.13: "appear to converge": the convergence should be better studied to assess if there is convergence with grid resolution.

p.2 l.4: "inherent dynamic instabilities associated with marine-based ice sheets". The marine instability should be briefly explained in a couple sentences.

p.4 l.28-30: It should be stated here that errors in full-Stokes models and lower-order approximations should be accurately characterized to assess if the differences are within the error margins or if these models do lead to different results.

p.5 l.2: "Finite Element Methods" → "the Finite Element Method"

p.5 l.30: What about the vertical resolution used in both models?

p.6 l.4: "the prognostic, basal sliding perturbation experiment (P75S and P75R)": is that one or two experiments?

p.6 l.20: What resolution is used for the diagnostic experiment?

p.7 l.21: What is the vertical resolution?

p.6 l.11: Replace "S", "R", ... by the experiment name.

p.8 l.15: Why are the grid different for these experiments? If it is possible to use the same grid for experiments Stnd and P75D, it should also be possible to use a similar grid for the other experiments (especially as they are much shorter than the Stnd experiment).

p.8 l.21: "the GL improves": do you expect the grounding line to come back to its initial position on such a short time scale?

p.8 l.27-29: "We attribute ..." I attribute it to the coarse grid used in the across flow direction.

p.9 l.13-23: This paragraph should be moved to section 3.

p.9 l.20: "basis functions": a regular grid is used so the basis functions should have similar properties.

p.10 l.32-33: I don't agree with the conclusion that "two or more full-Stokes models should first conduct their own intercomparion". I think that just like any numerical model, they should make sure that their results are not grid dependent and provide an error associated with their results.

p.11 l.4: What does "a more direct comparison between models" mean? The two models are already sharing lots of parameters, even the same grid. This should be detailed and not just mentioned.

p.14 Tab.2: There are no Stnd results ($x_{GO}$) provided for the 40 and 80 elements in the $y$ direction. So what are the initial states used for the P75S experiments in these cases?

p.15 Fig.1: The friction coefficient used along the profile is not totally clear, especially for Profile 2 (it looks like a straight line, which is surprising).

Fig.4, 5 and 6: Thin (or thick) curves should be used consistently for the 4 panels to represent the same FELIX-S results. Also consider adding letters (a to d) for the

[Figure]

panels, as mentioned in the captions.

Fig.7: How can an absolute value be negative (right panel)?

**4  References**

G. Durand, O. Gagliardini, T. Zwinger, E. Le Meur, and R. Hindmarsh, Full Stokes modeling of marine ice sheets: influence of the grid size, Annals of Glaciology, 50(52), 2009.

J. Feldmann, T. Albrecht, C. Khroulev, P. F., and A. Levermann, Resolution- dependent performance of grounding line motion in a shallow model compared with a full-Stokes model according to the MISMIP3d intercomparison, Journal of Glaciology, 60(220), doi:10.3189/2014JoG13J093, 2014.

O. Gagliardini, J. Brondex, F. Gillet-Chaulet, L. Tavard, P. V., and D. G., Brief communication: Impact of mesh resolution for MISMIP and MISMIP3d experiments using Elmer/Ice, The Cryosphere, 10, doi:10.5194/tc-10-307-2016, 2016.

F. Pattyn, et al., Results of the Marine Ice Sheet Model Intercomparison Project, MISMIP, The Cryosphere, 6(3), doi:10.5194/tc-6-573-2012, 2012.

F. Pattyn, et al., Grounding-line migration in plan-view marine ice-sheet models: re- sults of the ice2sea MISMIP3d intercomparison, Journal of Glaciology, 59(215), doi:10.3189/2013JoG12J129, 2013.

H. Seroussi, M. Morlighem, E. Larour, E. Rignot, and A. Khazendar, Hydro- static grounding line parameterization in ice sheet models, The Cryosphere, 8(6), doi:10.5194/tc-8-2075-2014, 2014.

---

## Referee Comment (RC2) · S. L. Cornford (Referee) · 13 Apr 2016

This paper presents results for the MISMIP3d experiments for the recent finite-element Stokes flow model FELIX-S and compares them with revised results from the well known Elmer/Ice model, (that have already been published in TC by Gagliardini 2016 et al). It is good to see a second high resolution Stokes model performing these prognostic experiments, and it looks as though the two models agree to within the published truncation error of Elmer/Ice, and indeed, seem to have similar truncation error. Notably, both models a steady state grounding line position about 80 km upstream of the SSA models, as seen for Elmer/Ice in the original MISMIP3d results and as the truncation error at an along-flow mesh spacing of 50m is around 5km, the paper adds weight

to the view that this difference is due to the more complete physics in Stokes model than numerical error.

I have one major comment, and some specific comments

Major Comment

It seems that the difference between the models is entirely truncation error, ie both models appear to implement the same physics. At several places the paper suggest that the difference between the models tells us what the error in those models is. That's not quite right – it tells us what what the larger truncation error of the two models is, but not which model has that error. Imagine that Elmer/Ice had a 5km error in the GL position and Felix-S had 100 m error (I know that's not true – this is just an illustration). Would the Felix-S developers be happy to accept that the error in their model was 5 km on that basis?

It is not enough to compare two Stokes models: we need to compare models that have known truncation error. On top of that, if we know the truncation error (and that is the major issue), then the comparison is much less important than this paper suggests in several places including the abstract and conclusion. So while it is certainly desirable for future MISMIP type exercises to include multiple Stokes models, the proposal that it be a requirement, is, I think, an opinion rather than a fact, and should not appear in the published paper.

There should be a convergence study of the Stnd experiment run by FELIX-S with respect to along flow mesh spacing dx. Other papers (e.g Seroussi 2015 ) make it clear that the Stnd experiment is harder to get right than the reversibility. Gagliardini 2016 included such a study and I think this paper should too. That is, run the Stnd experiment at dx = 100m , dx =200 m, dx = 400 m. etc in addition to dx = 50m ( dx = 25 might be too expensive). That should allow an estimate of the truncation error as a function of dx (it will presumably be about 100 dx, as it is for Elmer/Ice) – this is already hinted at by the 5km difference in the two dx = 50m experiments mentioned on page 8.

Minor Comments

Fig 7: The caption talks about an absolute value of x (ie |x|) but there are negative values on the plot. Also, what does the least-squares line tells us about convergence? I would expect $|x| \sim dx = 1/N_y$. I suggest plotting the |x| differences on a log-log scale.

P8, L20; "reversibility" is not better just because the GL is closer to its initial point after 100 years. It takes $\sim 10\hat{}4$ years to see the return to steady state. What's important here is the convergence of the FELIX-S and Elmer/Ice solutions within themselves and with each other.

P9, L25: likelihood has a specific meaning. Maybe use 'tendency' or something else?

P11, L2, and elsewhere. Uncertainty? I would say error, numerical error, or some such. Uncertainty is quite a broad term suggesting probability , but here we know that differences are due to (deterministic) error.

P11, L4: Future efforts. I don't really see the value of comparing identical treatments. I guess it might expose bugs, or differences due to solution method (as opposed to discretization methods), but it's not usual to publish that sort of thing (even if bugs are a common enough occurrence)

---

## Referee Comment (RC3) · Anonymous Referee #3 · 29 Jun 2016

This manuscript aims at a numerical comparison of a new computational code FELIX-S to the already published computational code Elmer/Ice. Both codes ("models") are based on the same equations: the Stokes equations with grounding-line type boundary conditions. The present code FELIX-S is based on the Hood-Taylor finite elements (second order) while Elmer/Ice is based on the mini-finite element (first order). The dynamic grounding line conditions (friction conditions on the left, floating conditions on the right plus a contact threshold condition) are implemented slightly differently. The two different implementations are presented. The numerical comparisons are made both for prognostic and diagnostic MISMIP3d benchmarks. The two computational codes give reasonably similar results in all cases; the observed discrepancies are very

likely due to the difference in the implementation of the friction boundary condition and the accuracy difference of the two finite elements used. In this sense, this new Stokes code ("model") is interesting for the glaciology community since it proposes an additional computational code solving the Stokes equations with friction and dynamic grounding line.

But the crucial scientific questions and issues are not addressed; in particular the convergence of these codes when refining the mesh in the grounding line vicinity. Such a convergence study is the first step to assess any numerical model before interpreting the results in terms of physics. The close agreement in model outputs between the present two codes demonstrate that they probably do not contain any programming bugs; but it does not demonstrate the validity and reliability of their results in terms of modelling since the two models are the same. (Note that in the manuscript the terminology "model" is inadequately employed since the two codes consider exactly the same physical model solved by very similar numerical methods). These two Stokes models could give reference solutions for the crucial and difficult grounding line problem, in particular when comparing to asymptotic shallow models (SSA), if their assessments would have been complete. To my knowledge that is not fully the case yet, since the crucial issue of convergence seems to remain.

In short, this manuscript is a good description of a new and additional computational code solving the complete Stokes system; it is nicely compared to the Elmer/Ice code. But this manuscript does not address the question of the grounding line modelling nor does it answer the crucial issue of non convergent models.

This is the reason why this manuscript version cannot be considered as a research publication; it may be suitable for the Geoscientific Model Development journal.

---

## Author Comment (AC1) · 8 Jul 2016

We disagree with the reviewer that the contribution is not novel – while grounding lines within Stokes models has been investigated for many years now, it has only been investigated in any amount of detail within a single model, Elmer/Ice. That same model has been used extensively as a metric for accuracy in community-wide, model intercomparison exercises, including two widely used and widely cited intercomparisons focused on marine ice sheet and grounding line dynamics. A stated goal of our contribution is to demonstrate how similar or different the solutions are from two Stokes models, which are based on significantly different, but equally valid implementations, in order to provide additional confidence in the use of Stokes models as an accuracy metric in past and future intercomparison exercises. As many of the results from these intercomparisons (and related papers) have been published in *The Cryosphere*, we are confident that our results will be of sufficient interest to warrant their publication therein.

*Second, I found the abstract in particular to be misleading, and to put emphasis on the "wrong" solutions to real questions. As mentioned by the authors, "as grid resolution increases the grounding line positions for FELIX-S and Elmer/Ice appear to converge", which suggests that full-Stokes models (just like any numerical model) should do a convergence study to assess the impact of grid resolution on their results.*

We agree with the reviewer regarding the issue of convergence with respect to grid resolution (see additional discussion on this topic below). We add the "appear" qualifier here because it is not obvious that solutions from two different models can be shown to converge, in the formal sense, with increasing grid resolution. Further, with limited computing resources, such a study may not actually be feasible. What is feasible, and what we clearly demonstrate here, is that when the models are run at identical grid resolutions, and as that resolution increases, the model solutions become more and more similar to one another. While convergence of solutions between the two models is implied but not proven, we strongly believe that this result is worthy of publication in order to provide additional confidence in the use of Stokes models as accuracy metrics in intercomparison exercises. We have added additional figures and discussion in the revised manuscript to show that (1) the FELIX-S model is convergent with increasing

grid resolution (last paragraph of section 5.2 and new Figure 4), and (2) that the differences between the FELIX-S and Elmer/Ice models at their finest resolutions are generally near or below previously published estimates for the Elmer/Ice truncation error.

*The authors on the opposite suggest that "future model intercomparisons using full-Stokes models as a metric should include more than one model, to provide both additional confidence in the results from full-Stokes models and a measure of their uncertainty". Comparing two different full-Stokes models will not allow quantifying the error in any of these mod- els.*

Our intention is not to quantify the error in the Stokes models or between Stokes models and any other models. Our intention, which is clearly stated, is to provide additional confidence in the use of Stokes models by showing that two very different Stokes models give very similar solutions when model resolution is sufficiently high. Implicit is the understanding that any single model (Stokes or otherwise) is responsible for confirming numerical convergence with increasing grid resolution (see also next response below). In the revised version of the manuscript, we have also removed the suggestion that future intercomparison projects include more than one Stokes model.

*The only way to assess the error of each model is to do a convergence study, with varying grid resolution, to see if the results are converging, and if so, to quantify the error associated to the discretization. To my knowledge, this has never been properly done in three-dimensional models, and needs to be done instead of writing numerous papers on variations along the same subject (comparison of friction treatment around the grounding line, comparison of several full-Stokes models, ...).*

In fact, both models discussed here have previously published studies on solution convergence with increasing grid resolution. For the Elmer/Ice model, this has been discussed in detail in Gagliardini et al. (2013), section 7.1 and Gagliardini et al. (2016; see also Supp. Inf.). For the FELIX-S model, this has been discussed in detail in Leng et al. (2012), section 4.1 and Leng et al. (2013) section 4.2. This was not made explicit in the original version of the manuscript and we have added additional discussion to that end at the beginning of Section 3. As noted above, we have also added additional discussion and a new figure to address convergence of the FELIX-S model for the Stnd experiment of the MISMIP3d experiment suite.

*Finally, the results presented in this paper do not present major new results, as the conclusions are somewhat similar those of Gagliardini et al. (2016), which already discuss the impact of different treatment of basal friction around the grounding line.*

The fact that the results and conclusions of this paper are similar to those in Gagliardini et al. (2016) is precisely the point. The "major new result" is that we know have additional confidence when using Elmer/Ice (or FELIX-S) as a metric for high-fidelity model solutions in benchmarking exercises. Imagine the case where only one or the other of the Stokes models discussed here is to be used as a metric for solution accuracy in a model intercomparison (e.g., against which other reduced-order model solutions are to be compared). The community is already aware of solution errors resulting from underresolution. But both Gagliardini et al. (2016) and the present work demonstrate that seemingly arbitrary differences in implementation can also result in significant *differences* in model solutions. We emphasize *differences* here because it is not clear these can be characterized as errors – they result from a choice in implementation method. One or another model alone cannot be used to quantify such *differences* but multiple models can. And that characterization provides some estimate for how much confidence can be attached to the results from a single Stokes model.

*As errors in full-Stokes models start to be better estimated, and lower order approximations models also improved grounding line treatment during the past few years (Feldmann et al., 2014; Seroussi et al., 2014), a more interesting study would be to reassess the difference between full-Stokes and lower order approximations based on all these new data.*

We certainly agree with the reviewer on this point. However, this is an entirely different contribution than the present one. Again, our primary (and stated) goal here is to provide additional confidence in the use of Stokes models as accuracy metrics in model intercomparison exercises.

**Response to Specific Comments**

*The convergence or absence or convergence with grid resolution should be rigorously assessed using a convergence study instead of providing statements like "appear to converge" in the abstract.*

As noted above, we have revised the manuscript to point to previous work where formal convergence studies have been conducted for both models, and to add discussion on a convergence study for the FELIX-S model applied to one of the standard MISMIP3d experiments. The differences we are trying to quantify here result not strictly from grid resolution but from different implementation choices, which cannot necessarily be characterized as being relatively better or more correct in one model or the other.

*The description of the friction coefficient in section 3 is not clear. Figure 1 is also confusing, especially the friction coefficient along profile 2. This seems to be a straight line while there is a node in the middle of the profile. The details provided in the discussion should be moved in this section (p.5 l.15-16).*

See response to this same concern in the technical comment section below.

*p.8 l.3-7: For which model is there a 5 km difference?*

This statement was referring to FELIX-S. We've clarified this in revision.

*This paragraph also mentions that this distance is expected to decrease with increased grid resolution. Why not do these runs and provide an accurate answer? These positions should be the same for a model that has converged with grid resolution. So here again,*

*this seems to suggest that the results presented have not converged.*

A more complete convergence study of the sort suggested was done and is reported on in Gagliardini et al. (2016). Computing resource limitations prevented us from doing as exhaustive of a study here. We have re-worded this section slightly in order to make it clearer that we are speculating that FELIX-S would show similar results if we had done a similar study.

*How long is the steady-state run for? Could this difference be partly explained because the steady-state should be run longer to fully converge?*

For the steady-state Stnd experiments, FELIX-S simulations are started from the same initial condition used by Elmer/Ice, which is based on the boundary-layer theory solution of Schoof (2007). In this sense, it should already be close to equilibrium. We then integrate forward in time for ~1300 years (or ~1500 years, depending on whether starting from an advanced or retreated state). The criterion for an equilibrium configuration is the same as that used by Elmer/Ice (i.e., that the relative rate of volume change is below $10^{-5}$). Thus, we are confident that the difference in steady-state GL positions is *not* due to the models not being fully converged and in equilibrium. The discussion of the Stnd experiment has been updated to provide more explicit information on these details.

*p.8 l.22-23: "the agreement between FELIX-S and Elmer/Ice increases for all of Elmer/Ice GL implementations". This seems like another sign that the differences in the results are caused by the non-converged aspect of the results presented.*

We don't disagree with the reviewer on this point, and we discuss this in detail in the text already. At relatively coarser resolutions, the implementation choices that we discuss in detail have a relatively larger effect on model solution differences. At relatively finer resolution, these implementation differences are minimized, and the model solutions become more similar. We believe this is discussed and explained in adequate detail in the current version of the manuscript.

*p.1 l.2 (and other places in the text): full Stokes is commonly used in the literature, so there is no need to use "full" Stokes.*

In fact, the use of the term "full Stokes" is unique to the glacier / ice sheet modeling community. We apply the quotes in the first use as a way of indicating that this is somewhat mushy terminology. As suggested, we've removed the quotes after the first use.

*p.1 l.13: "appear to converge": the convergence should be better studied to assess if there is convergence with grid resolution.*

See multiple responses to this same concern above.

*p.2 l.4: "inherent dynamic instabilities associated with marine-based ice sheets". The marine instability should be briefly explained in a couple sentences.*

The marine ice sheet instability has been discussed in great detail in numerous previous papers (several of which have been added to the introductory section here as standard references) and in the original model intercomparison paper this work stems from. Moreover, readers interested in this contribution will already be familiar with the marine ice sheet instability. Therefore, we do not think it necessary to review that topic in additional detail.

**Technical Comments**

*p.4 l.28-30: It should be stated here that errors in full-Stokes models and lower-order approximations should be accurately characterized to assess if the differences are within the error margins or if these models do lead to different results.*

We've added some additional discussion to this effect in the multiple sections of the revised paper. In particular, we point out in a number of different areas that, at highest resolution, the differences between Elmer/Ice and FELIX-S are within the estimated truncation error of the Elmer/Ice model.

*p.5 l.2: "Finite Element Methods" → "the Finite Element Method"*

Changed.

*p.5 l.30: What about the vertical resolution used in both models?*

Information about the vertical resolution has been added to section 3.

*p.6 l.4: "the prognostic, basal sliding perturbation experiment (P75S and P75R)": is that one or two experiments?*

We leave the description as in the original MISMIP3d paper, where it is described as a single experiment but with two parts (the "S" and "R" parts). To be clearer, "experiment" has been changed to "experiments".

*p.6 l.20: What resolution is used for the diagnostic experiment?*

Additional information on the meshes used for all experiments has been added Section 3.

*p.7 l.21: What is the vertical resolution?*

Information about the vertical resolution has been added to section 3.

*p.6 l.11: Replace "S", "R", ... by the experiment name.*

Presumably p.8 is meant. Changed as suggested.

*p.8 l.15: Why are the grid different for these experiments? If it is possible to use the same grid for experiments Stnd and P75D, it should also be possible to use a similar grid for*

*the other experiments (especially as they are much shorter than the Stnd experiment).*

As stated in the first paragraph of Section 5.3, for the P75S and P75R experiments the regionally refined portion of the mesh (for both models) is based on the steady-state grounding line at the end of the Stnd experiment. Because these locations are slightly different for the two models, the refinement region is also slightly different, and thus nodal coordinates are not identical as they are for the P75D and Stnd experiments.

*p.8 l.21: "the GL improves": do you expect the grounding line to come back to its initial position on such a short time scale?*

No. Based on previous work, we expect that >>100 yrs is required for this. We've added a parenthetical note and reference to the text to clarify this.

*p.8 l.27-29: "We attribute ..." I attribute it to the coarse grid used in the across flow direction.*

When we say differences, it is implied that we are talking about differences when using the same across and along-flow grid resolutions. If the models are using the same grid resolution, then presumably the differences are due to something *other than* the grid resolution.

*p.9 l.13-23: This paragraph should be moved to section 3.*

This is a matter of style in composing the manuscript and we disagree with the reviewer. If all of this information were given in section 3, it would not be at all clear to the reader what the context for so much detail was. Whereas, the context is much more clear in the discussion section.

*p.9 l.20: "basis functions": a regular grid is used so the basis functions should have similar properties.*

If the both grids are quadrilateral (square element faces in map view), then yes. But FELIX-S uses tetrahedral elements (triangular element faces in map view, with two acute and one right angle). The basis functions that contribute to any node will depend on the shape and orientation of the surrounding triangles. Further, FELIX-S uses 2nd-order accurate basis functions whereas Elmer/Ice uses linear basis functions.

*p.10 l.32-33: I don't agree with the conclusion that "two or more full-Stokes models should first conduct their own intercomparion". I think that just like any numerical model, they should make sure that their results are not grid dependent and provide an error associated with their results.*

Please see responses and discussion above.

*p.11 l.4: What does "a more direct comparison between models" mean? The two models are already sharing lots of parameters, even the same grid. This should be detailed and not just mentioned.*

The details the reviewer requests are given in the lines following this statement ("This would include …").

*p.14 Tab.2: There are no Stnd results ($x_{GO}$) provided for the 40 and 80 elements in the y direction. So what are the initial states used for the P75S experiments in these cases?*

The Stnd experiment is invariant in the across-flow direction (the initial conditions as well as the model solutions). Thus, adding additional resolution to the across-flow direction has no impact on the results of the Stnd experiment. This should be clear from the description of the experiments given in section 4, but we have also added a sentence to the caption for Table 2 to clarify this.

*p.15 Fig.1: The friction coefficient used along the profile is not totally clear, especially for Profile 2 (it looks like a straight line, which is surprising).*

This is because the friction coefficient used in these experiments is uniform and constant (see Table 1). The reasons why Elmer/Ice and FELIX-S "see" the friction coefficient at the nodes slightly differently is discussed in detail in the text. In the more general case of spatially varying friction coefficient, $C=C(x,y)$, and the friction profile for Elmer/Ice would also appear more complicated.

*Fig.4, 5 and 6: Thin (or thick) curves should be used consistently for the 4 panels to represent the same FELIX-S results. Also consider adding letters (a to d) for the panels, as mentioned in the captions.*

We've improved the different panels in Figures 4-6 (black bold is now always FELIX-S results, as suggested). The panels do contain letters, in the lower right corners. In revision, we've made them bold so they are easier to see.

*Fig.7: How can an absolute value be negative (right panel)?*

In response to similar comments from reviewer no. 2, we have completely revised this figure, the figure caption, and the related discussion in the text.

**References**

Gagliardini, O., and Coauthors, 2013: Capabilities and performance of Elmer/Ice, a new-generation ice sheet model. *Geosci. Model Dev*, **6**, 1299–1318, doi:10.5194/gmd-6-1299-2013.

Gagliardini, O., J. Brondex, F. gillet-Chaulet, L. Tavard, V. Peyaud, and G. Durand, 2016: Brief communication: Impact of mesh resolution for MISMIP and MISMIP3d experiments using Elmer/Ice. *The Cryosphere*, **10**, 307–312, doi:10.5194/tc-10-307-2016-supplement.

Leng, W., L. Ju, M. Gunzburger, S. Price, and T. Ringler, 2012: A parallel high-order

accurate finite element nonlinear Stokes ice sheet model and benchmark experiments. *J. Geophys. Res*, **117**, doi:10.1029/2011JF001962.

Leng, W., L. Ju, M. Gunzburger, and S. Price, 2013: Manufactured solutions and the verification of three-dimensional Stokes ice-sheet models. *The Cryosphere*, **7**, 19–29, doi:10.5194/tc-7-19-2013.

Schoof, C.: Marine ice-sheet dynamics. Part 1. The case of rapid sliding, *Journal of Fluid Mechanics*, **573**, 27, doi:10.1017/S0022112006003570, 2007b.

---

## Author Comment (AC2) · 8 Jul 2016

We thank the reviewer for his comments and for pointing out the model similarities in the context of the published Elmer/Ice truncation error. In the original version of our manuscript, we did not explicitly point this out. In our edited version, we have made it more explicit in several places that, when run at their highest grid resolutions the two models discussed here show agreement to within the published Elmer/Ice truncation error of 2-3 km.

**Response to Major Comments**

*It seems that the difference between the models is entirely truncation error, ie both models appear to implement the same physics. At several places the paper suggest that the difference between the models tells us what the error in those models is. That's not quite right – it tells us what the larger truncation error of the two models is, but not which model has that error. Imagine that Elmer/Ice had a 5km error in the GL position and Felix-S had 100 m error (I know that's not true – this is just an illustration). Would the Felix-S developers be happy to accept that the error in their model was 5 km on that basis?*

*It is not enough to compare two Stokes models: we need to compare models that have known truncation error. On top of that, if we know the truncation error (and that is the major issue), then the comparison is much less important than this paper suggests in several places including the abstract and conclusion. So while it is certainly desirable for future MISMIP type exercises to include multiple Stokes models, the proposal that it be a requirement, is, I think, an opinion rather than a fact, and should not appear in the published paper.*

We take the reviewers point, and have updated the relevant discussion in the abstract, the discussion, and conclusions sections of the paper. We now state explicitly that, at high resolution, the differences between the two models appear to largely fall within the previously published estimates for the truncation error of Elmer/Ice. While we agree that this diminishes the requirement that multiple Stokes models participate in future intercomparison exercises, we do include a strongly worded opinion that this should still be the case, simply as a way of confirming that in future studies.

*There should be a convergence study of the Stnd experiment run by FELIX-S with respect to along flow mesh spacing dx. Other papers (e.g Seroussi 2015 ) make it clear that the Stnd experiment is harder to get right than the reversibility. Gagliardini 2016 included such a study and I think this paper should too. That is, run the Stnd experiment at dx=100m,dx=200m,dx=400m. etc in addition to dx=50m(dx= 25 might be too expensive). That should allow an estimate of the truncation error as a function of dx (it will presumably be about 100 dx, as it is for Elmer/Ice) – this is already hinted at by the 5km difference in the two dx = 50m experiments mentioned on page 8.*

As suggested, we now report on the results of the grid convergence study for the Stnd experiment. There is a new paragraph discussing this at the end of section 5.2 and a new figure showing the results of the convergence experiment (new Figure 4).

**Response to Minor Comments**

*Fig 7: The caption talks about an absolute value of x (ie |x|) but there are negative values on the plot. Also, what does the least-squares line tells us about convergence? I would expect |x| ∼ dx = 1/N_y. I suggest plotting the |x| differences on a log-log scale.*

We have replaced the previous version of Figure 7 with a newer one in the spirit of the reviewer's suggestion.

*P8, L20; "reversibility" is not better just because the GL is closer to its initial point after 100 years. It takes ∼10ˆ4 years to see the return to steady state. What's important here is the convergence of the FELIX-S and Elmer/Ice solutions within themselves and with each other.*

We have added a clarifying comment to this effect in the revised version of the manuscript.

*P9, L25: likelihood has a specific meaning. Maybe use 'tendency' or something else?*

In the revised version of the manuscript, the relevant sentence has been changed to: "This in turn *will make it more likely* that neighboring nodes may also eventually ground."

*P11, L2, and elsewhere. Uncertainty? I would say error, numerical error, or some such. Uncertainty is quite a broad term suggesting probability , but here we know that*

*differences are due to (deterministic) error.*

This part of the conclusions (formerly p.11) has been re-written. Elsewhere in the paper where we formerly used the word "uncertainty", we have changed it to "error", "model error", etc.

*P11, L4: Future efforts. I don't really see the value of comparing identical treatments. I guess it might expose bugs, or differences due to solution method (as opposed to discretization methods), but it's not usual to publish that sort of thing (even if bugs are a common enough occurrence).*

We have removed this particular section of the final paragraph. Now the "future efforts" discussion focuses on trying to identify if the different methods used by FELIX-S lead to different or similar truncation errors relative to those found for Elmer/Ice.

---

## Author Comment (AC3) · 8 Jul 2016

**Response to Major Comments**

*This manuscript aims at a numerical comparison of a new computational code FELIX- S to the already published computational code Elmer/Ice. Both codes ("models") are based on the same equations: the Stokes equations with grounding-line type boundary conditions. The present code FELIX-S is based on the Hood-Taylor finite elements (second order) while Elmer/Ice is based on the mini-finite element (first order). The dynamic grounding line conditions (friction conditions on the left, floating conditions on the right plus a contact threshold condition) are implemented slightly differently. The two different implementations are presented. The numerical comparisons are made both for prognostic and diagnostic MISMIP3d benchmarks. The two computational codes give reasonably similar results in all cases; the observed discrepancies are very likely due to the difference in the implementation of the friction boundary condition and the accuracy difference of the two finite elements used. In this sense, this new Stokes code ("model") is interesting for the glaciology community since it proposes an additional computational code solving the Stokes equations with friction and dynamic grounding line.*

*But the crucial scientific questions and issues are not addressed; in particular the convergence of these codes when refining the mesh in the grounding line vicinity. Such a convergence study is the first step to assess any numerical model before interpreting the results in terms of physics. The close agreement in model outputs between the present two codes demonstrate that they probably do not contain any programming bugs; but it does not demonstrate the validity and reliability of their results in terms of modelling since the two models are the same. (Note that in the manuscript the terminology "model" is inadequately employed since the two codes consider exactly the same physical model solved by very similar numerical methods). These two Stokes models could give reference solutions for the crucial and difficult grounding line problem, in particular when comparing to asymptotic shallow models (SSA), if their assessments would have been complete. To my knowledge that is not fully the case yet, since the crucial issue of convergence seems to remain.*

Initially in response to reviewers 1 and 2, we have added a new section and two new figures, demonstrating (1) the convergence of the FELIX-S code with respect to along-flow grid resolution (for the Elmer/Ice code, convergence has been discussed and published in a number of previous papers, e.g. Durand et al. (2009) and Gagliardini et al. (2016)), and (2) that the results for Elmer/Ice and FELIX-S also converge with increasing resolution to within the previously published truncation error for the Elmer/Ice model. As in our discussion and response to reviewers 1 and 2, we feel that our revised manuscript addresses the concerns of reviewer 3 with respect to numerical convergence.

*In short, this manuscript is a good description of a new and additional computational*

*code solving the complete Stokes system; it is nicely compared to the Elmer/Ice code. But this manuscript does not address the question of the grounding line modelling nor does it answer the crucial issue of non convergent models. This is the reason why this manuscript version cannot be considered as a research publication; it may be suitable for the Geoscientific Model Development journal.*

As noted above, we feel that our revised manuscript now does address the issues of model convergence that are of concern to this reviewer. As for the last criticism, we disagree that this paper is not valid as a research contribution to *TC*. In our revised manuscript, we show that the two codes demonstrate convergent behavior, at least to within the level of truncation error found for the Elmer/Ice model (if the error were smaller for FELIX-S, it would not matter since the models can only be compared to within the larger of the two errors). This provides a strong argument for continuing the practice of treating Stokes model solutions as an accuracy metric in ongoing and future model intercomparison exercises, a conclusion that should be of wide interest to the ice sheet modeling community and the audience of *TC*. We also note that recently published papers in *TC* (Gagliardini et al., 2016) are similar in nature in that they discuss and explore different choices in model implementation and the impacts they have on modeled GL position. These same choices will have significant impacts on model solutions as part of ongoing community intercomparison projects (e.g., MISMIP+ and MISOMIP, Asay-Davis et al. (2016)) and realistic simulations conducted for research purposes (e.g., estimation of ice flux and GL evolution in basin-scale experiments of Antarctica and Greenland). Therefor, we feel that our results will be of great interest to many regular readers of *TC*.

**References**

Durand, G., O. gagliardini, B. De Fleurian, T. zwinger, and E. Le Meur, 2009: Marine ice sheet dynamics: Hysteresis and neutral equilibrium. *J Geophys Res-Earth*, **114**, –, doi:10.1029/2008JF001170.

Gagliardini, O., J. Brondex, F. gillet-Chaulet, L. Tavard, V. Peyaud, and G. Durand, 2016: Brief communication: Impact of mesh resolution for MISMIP and MISMIP3d experiments using Elmer/Ice. *The Cryosphere*, **10**, 307–312, doi:10.5194/tc-10-307-2016-supplement.

---

## Author Response (AR2)

**Point-by-Point response to the second set of reviews for "*A comparison of two Stokes ice sheet models applied to the Marine Ice Sheet Model Intercomparison Project for plan view models (MISMIP3d)*"**

The point-by-point discussion of changes made to the revised version of the manuscript is best followed by reviewing the response to each reviewer comment. Where a change to the original manuscript was made, it is clearly noted, including what that change was. If no change was made, we note this and present arguments for why we made this choice.

**Reviewer number 1**

*p.2 l.16-20: This statement is not justified. It is not clear why models would need comparison with Stokes models more than in the past. Furthermore, if Stokes models do not make any assumption in the stress tensor, there has not been any improvement in the treatment of grounding line with subgrid parameterization similar to what has been done for other approximations. This makes Stokes models more sensitive to grid resolution than other models.*

We respectfully disagree that this statement is not justified. It is, in fact, true that recent model intercomparison exercises (a number of them referenced here) have become too complex to allow for the use of analytical solutions as accuracy metrics. And, as a result, Stokes model solutions have been, and will likely continue to be, used for that purpose. We do agree with the argument that Stokes model solutions in the vicinity of the grounding line are very sensitive to grid resolution (as is clearly shown and discussed in detail here and in other papers referenced). However, that point is off topic relative to the discussion in this section of the introduction, which is recounting how Stokes models have been used / continue to be used in intercomparison exercises, providing an argument for why a comparison between independently developed Stokes models is a worthwhile endeavor.

*The manuscript remains too qualitative in several places. Adverbs like "slightly" (in section 5.2 in particular, but in several other place in the text) are used very often and should be replaced by quantitative measures (e.g. "slightly thinner ice (and hence floatation) occurring slightly farther inland relative to Elmer/Ice").*

We have read carefully through the manuscript and, wherever possible, made an effort to replace qualitative statements with a more quantitative counterpart. At the same time, we do not want to clutter the manuscript with an unnecessary level of detail (e.g., if the relevant metric for two comparative simulations is the GL position, and the difference in GL position are stated quantitatively, is it necessary / useful to also explicitly specify the value of the minor thickness and velocity differences that contribute to the GL differences?). Therefore, in our revised manuscript, we've tried to strike the appropriate balance between adequate versus excessive quantitative detail (and we are willing to further adjust this level of detail if necessary). This includes removing, wherever possible, unclear qualifying language like "slightly".

- intro, 2nd to last paragraph – have changed "very high resolution" to "high

resolution (e.g., 50 m along flow)"

- section 3, paragraph 2 – "slightly different masking schemes" → "different masking schemes"
- section 3, paragraph 3 – "information is used slightly differently" → "information is used differently"
- section 3, last paragraph – "nodal coordinates are very similar but not identical" → "nodal coordinates are not identical"
- section 5.1, last paragraph – removed two instances of the qualifier "slightly" (in "slightly larger" and "slightly different")
- section 5.2, first paragraph – "slightly different numerics" → "different numerics")
- section 5.2, second paragraph, last sentence – remove two instances of the qualifier "slightly" and add quantitative detail on the range of differences between the FELIX-S and Elmer/Ice equilib. GL positions
- section 5.2, third paragraph – remove two instances of the qualifier "slightly"; add detail on the magnitude of the velocity difference between FELIX-S and Elmer/Ice
- section 5.2, 2$^{nd}$ to last paragraph – add text to clarify that 50 m along-flow resolution applies to the region in the vicinity of the GL (note that this change has also been made elsewhere in the manuscript); remove "seemingly" in last sentence
- section 5.2, last paragraph – added detail regarding high-resolution in the vicinity of GL (that is, it is generated through a geometric progression, as in previous related work)
- section 5.3, first paragraph – remove two instances of the qualifier "slightly"
- section 6, fifth paragraph – remove qualifier "slightly"
- section 7, first paragraph – remove two instances of the qualifier "slightly"

*Section 5.3 explains the importance of grid resolution in both directions in getting similar results between Elmer/Ice and FELIX-S, and I think this statement should be added to the abstract along [with?] the other conclusions about experiments P75S and P75R.*

In the abstract, we have clarified that convergence of results from the two models is a function of horizontal (both along- and across-flow) mesh resolution.

Technical comments

*p.5 l.9: "not clear" → "no clear"*

It seems that this sentence, "… there are not clear arguments …" is grammatically correct as written. However, we see that the suggested change, "… there are no clear arguments …", is also probably equally correct. Therefor, we have left it as is for now, but are happy to change it to the reviewer's suggestion based on the recommendation of the editor.

*p.11 l.29: "then" → "than"*

This correction has been made in the revised manuscript.

**Reviewer number 2**

*The convergence study suggest an error estimate in GL position of ~300 m at dx_min = 50m. But the difference between 'grown' and 'shrunk' GLs at dx_min = 50 is ~5 km . That, together with the neither linear nor quadratic form of the Richardson estimates tells us that the FELIX-S is not in the asymptotic regime (nor is Elmer/Ice), either because a finer resolution is needed at the GL, or the region of fine resolution needs to be bigger. However, I don't believe that anyone else has achieved better results than this. Perhaps the lesson is 'estimate the error in as many ways as possible and take the largest number'*

While it is not entirely clear if the referee is asking for it, we have added some additional discussion of this topic in the discussion section of the manuscript. This includes explicitly discussing why our grid convergence study is more of a "quasi-convergence" study (e.g., because we do not double the grid resolution everywhere in the domain at each step of the convergence study, but rather only along-flow and in the region of the GL, as required to manage computational costs).

*In the abstract, we have 'at a particular resolution, the span of grounding lines positions provides one estimate for model error ... More importantly we show that the grounding line positions appear to converge to within the estimated truncation error of Elmer/Ice'. I think this needs to be rephrased, because you always need the convergence study (different experiments result in different truncation errors), and so the model difference should never be needed as an error estimate - but might provide a useful complementary estimate where the convergence study cannot be carried to the asymptotic regime.*

The latter sentiment – that the range of GL positions from multiple models could serve as an estimate for uncertainty in the case where convergence studies were not practical – is exactly the idea that we were trying to communicate. We have restructured / rephrased the abstract to clarify both this and the other point made by the reviewer. We have also added / changed a few sentences in the conclusions in order to clarify this point.

*I noticed that the other two reviewers seemed to think this was more of a GMD paper than a TC paper. I don't fully agree - the paper could be equally at home in GMD, but we do want glaciologists, rather than model developers, to understand the limitations of the models they choose. So I see no reason why this discussion cannot be in TC, especially since parts of it (e.g the Gagliardini 2016 paper) already are. There was also a question of novelty. There I think that, since we need results to be independently reproduced before we believe them, there must be some avenue open to publish reproduced results, especially the first reproduced results.*

We fully agree with the referee and appreciate his support for publication of this work in *The Cryosphere*.

Specific Comments

*P6, L30 : 'very good agreement' -> close agreement, or something that does not involve 'very'*

The text here has been changed to "show good agreement", as suggested.

*P7, L3: 'the horizontal velocity', u -> ' the x-component of the horizontal velocity', u*

The suggested change has been made.

*P7, L24 '...superior to the other': and in any case should vanish as dx -> 0*

A concluding sentence to this effect has been added to this section.

*P7, L26 : 'very' ...*

We've re-written this to be more precise. The sentence now reads, "… results demonstrate that the model velocities are within several percent of one another when using identical nodal-mesh coordinates …".

*P7, L32: (50 m in the vicinity of the GL and 2500 m, respectively) what about the x-resolution far from the GL?*

The procedure for defining the Stnd experiment mesh used by FELIX-S is similar to that used by Elmer/Ice (described in section 3.6 of Durand et al., 2009). Moving away from the ~30 km wide region of 50 m resolution near the grounding line, along-flow mesh resolution increases linearly to several 10's of km based on a geometric progression. An additional sentence with this description has been added to the 2$^{nd}$ paragraph of section 5.2.

*P7,L34 'in equlibrium ... based on' -> 'close to equilibrium… according to'*

The suggested change has been made.

*P8 L3 'more retreated' -> 'further upstream'*

The suggested change has been made (we chose "farther" rather than "further", as the latter is sometimes used as a modifier for time rather than space).

*P8 L35 solution error -> estimates of solution error*

The relevant part of the sentence in question is, "Figure 4 shows the Richardson estimate for the solution error …", from which we think it is clear that we are referring to an error estimate rather than the actual error.

*P9: L1 'clearly convergent' -> seemingly convergent*

The suggested change has been made.

*P11, L5 'gradients in ice sheet geometry' -> differences in ice sheet geometry between nodes*

The suggested change has been made.

**A comparison of two Stokes ice sheet models applied to the Marine Ice Sheet Model Intercomparison Project for plan view models (MISMIP3d)**

Tong Zhang[1,3,4], Stephen Price[2], Lili Ju[4], Wei Leng[5], Julien Brondex[6], Gaël Durand[6], and Olivier Gagliardini[6]

[1]State Key Laboratory of Severe Weather (LASW), Chinese Academy of Meteorological Sciences, Beijing, China
[2]Fluid Dynamics and Solid Mechanics Group, Los Alamos National Laboratory, Los Alamos, NM, USA
[3]State Key Laboratory of Cryospheric Sciences, Chinese Academy of Sciences, Lanzhou, China
[4]Department of Mathematics and Interdisciplinary Mathematics Institute, University of South Carolina, Columbia, SC, USA
[5]State Key Laboratory of Scientific and Engineering Computing, Chinese Academy of Sciences, Beijing, China
[6]Université Grenoble Alpes, CNRS, IRD, IGE, F-38000 Grenoble, France

*Correspondence to:* Stephen Price (sprice@lanl.gov)

**Abstract.**

We present a comparison of the numerics and simulation results for two "full" Stokes ice sheet models, FELIX-S (Leng et al., 2012) and Elmer/Ice (Gagliardini et al., 2013). The models are applied to the Marine Ice Sheet Model Intercomparison Project for planview models (MISMIP3D). For the  diagnostic experiment (P75D) the two models give  similar results

5 ($<2\%$ difference with respect to along-flow velocities) when using identical geometries and computational meshes, which we interpret as an indication of inherent consistencies and similarities between the two models. For the Stnd, P75S, and P75R  prognostic experiments, we find that FELIX-S (Elmer/Ice) grounding lines are relatively more retreated (advanced), results that are consistent with minor differences observed in the diagnostic experiment results and  that we show to be due to different choices in the implementation of basal boundary conditions  in the two models.

10 ~~on current understanding, neither set of implementations can be argued to be more or less favorable. In this case, we propose that, at a particular resolution, the span of 
[revised manuscript text omitted]

---

## Author Response (AR3)

Dear Hilmar,

We have made the most recent set of revisions you requested, which include:

1) explicit discussion of the equilibrium ice thickness at the grounding line for the Stnd experiment
2) explicit discussion of the location of the equilibrium grounding line relative to the floatation point for the Stnd experiment
3) the addition of a new figure showing the modeled and floatation surfaces near the grounding line for the Stnd experiment (also useful for demonstrating close agreement with Elmer/Ice, published in previous work by Durand et al. (JGR, 2009)
4) explicit mention that, during the P75S/R experiments, the groundling line thickness varies by less than 2% of its equilibrium value in the Stnd experiment

While we have attempted to acknowledge yourself and the referees in the updated version of the acknowledgements, it was not entirely clear to us how many referees there were in total. By my count, there were 3 anonymous referees (2 on the first round of revisions, one on the second) plus Steph Cornford. Is that correct?

Thanks for you hard work in improving this manuscript and helping us (hopefully!) see it through to publication.

Sincerely,

Steve Price and Tong Zhang

[revised manuscript text omitted]